# PV power modelling using solar radiation from ground-based measurements and CAMS: Assessing the diffuse component related uncertainties leveraging the Global Solar Energy Estimator (GSEE)

Nikolaos Papadimitriou[1,2], Ilias Fountoulakis[2], Antonis Gkikas[2], Kyriakoula Papachristopoulou[3], Andreas Kazantzidis[1], Stelios Kazadzis[3], Stefan Pfenninger[4], John Kapsomenakis[2], Kostas Eleftheratos[5,6], Athanassios A. Argiriou[1], Lionel Doppler[7], and Christos S. Zerefos[2,6,8,9]

[1]Department of Physics, University of Patras, 26500 Patras, Greece

[2]Research Centre for Atmospheric Physics and Climatology, Academy of Athens, 11521 Athens, Greece

[3]Physikalish-Meteorologisches Observatorium Davos, World Radiation Center (PMOD/WRC), 7260 Davos, Switzerland

[4]Faculty of Technology, Policy, and Management (TPM), Delft University of Technology, 2628 BX Delft, the Netherlands

[5]Department of Geology and Geoenvironment, National and Kapodistrian University of Athens, 15784 Athens, Greece

[6]Biomedical Research Foundation, Academy of Athens, 11527 Athens, Greece

[7]Deutscher Wetterdienst, Meteorologisches Observatorium Lindenberg – Richard Assman Observatorium (DWD, MOL-RAO), 15848 Lindenberg (Tauche), Germany

[8]Navarino Environmental Observatory (N.E.O.), 24001 Messinia, Greece

[9]Mariolopoulos-Kanaginis Foundation for the Environmental Sciences, 10675 Athens, Greece

Corresponding author: Nikolaos Papadimitriou (npapadimitriou@academyofathens.gr, Vasilissis Sofias 79, 11521 Athens, Greece)

**Abstract**

Accurate PV power production modelling requires precise knowledge of the distribution of solar irradiance among its direct and diffuse components. Since this information is rarely available, this

requirement can be addressed through the use of diffuse fraction models. In this study, we try to
quantify the errors in PV modelling when measurements of the diffuse solar irradiance are not
available. For this purpose, we use total and diffuse solar irradiance data obtained from ground-
based measurements of BSRN to simulate the PV electric output using GSEE. We have chosen five
sites in Europe and North Africa, with different prevailing conditions, where BSRN measurements are
available. GSEE incorporates an implementation of the Boland-Ridley-Lauret (BRL) diffuse fraction
model, along with a Climate Data Interface that enables simulations across different time scales.
We evaluate the capability of BRL in providing accurate estimations of the diffuse fraction under
diverse atmospheric conditions, with particular attention on the presence of clouds and aerosols
and assess the extent to which its associated errors propagate to energy production modelling.
Furthermore, we compare GSEE outputs when using CAMS radiation time-series as input instead of
ground-based measurements, to quantify the impact of the CAMS radiation product uncertainties in
PV modelling.
**Keywords**
Solar energy modelling; CAMS radiation; PV power modelling; aerosol; dust; solar radiation
**1. Introduction**
Decarbonizing the power sector in a sustainable manner is pivotal in the effort to mitigate climate
change (Edenhofer et al., 2011; Owusu & Asumadu-Sarkodie, 2016; IPCC, 2023) and the large-scale
deployment of Solar Energy offers significant prospects toward this objective (Kakran et al., 2024).
The available solar energy is a variable source, fluctuating across different timescales with a unique
solar-resource profile over individual locations (McMahan et al., 2013). Therefore, accurate solar
energy forecasting and resource assessment is crucial for minimizing the risk in selecting project
location, designing the appropriate solar-energy conversion technology, and integrating new sources
of solar based power generation into the electricity grid (Stoffel, 2013), while short-term, intra-hour
forecasts are critical for power plant operations, grid-balancing, real-time unit dispatching,
automatic generation control, and trading (Pedro et al., 2017).
Extending solar irradiance forecasting to derive PV power forecasts is essential in solar energy
applications. PV power modelling can be achieved through the following additional steps to solar
irradiance forecasting: (i) decomposing Global Horizontal Irradiance (GHI) into Diffuse Horizontal
Irradiance (DHI) and Direct Normal Irradiance (DNI); (ii) calculating the plane-of-array irradiance
incident on the surface of PV planes, whether static or mounted on a solar tracking system, and (iii)
simulating the PV power production primarily based on the in-plane irradiance (Blanc et al., 2017).
The scarcity of concurrent measurements of both solar irradiance components, coupled with the
complexity of their theoretical computation, has driven the development of numerous empirical
models for estimating the diffuse fraction (ratio of the diffuse-to-global solar radiation). A seminal
contribution in this area was made by Liu and Jordan (1960), who established a correlation between
the diffuse fraction and the clearness or cloudiness index (ratio of the global-to-extraterrestrial
radiation). These models predominantly rely on the clearness index as the principal predictor. They
are generally classified into single-predictor models and multi-predictor models, with the latter
incorporating additional astronomical variables for enhanced precision (Paulescu & Blaga, 2019).
Typically, these models are expressed as polynomial equations, ranging from the $1^{st}$ to the $4^{th}$ degree,
that link the diffuse fraction to the clearness index $DF = f(clearness\ index, *params)$ (Jacovides
et al., 2006). Boland et al. (2001) proposed the use of a logistic function instead of linear or simple
nonlinear functions of the clearness index.  Ridley et al. (2010) developed a multiple-predictor
logistic model, known as the Boland-Ridley-Lauret (BRL), which combines simplicity and reliable
performance across both the Northern and Southern Hemispheres. The BRL model extends Boland's
approach by adopting the hourly clearness index as the principal predictor and introducing the
following additional parameters: apparent solar time, daily clearness index, solar altitude, and a
measure of the persistence of global radiation level.  In the implementation of the BRL included in
the GSEE, the users set as input only the hourly clearness. Moreover, this implementation adopts the
updated parameters proposed by Lauret et al. (2013), which derived using data from nine worldwide
locations covering a variety of climates and environments across Europe, Africa, Australia and Asia.
While the existing models consider all-sky conditions, in solar energy modelling it is critical to focus
on cloud-free skies, where energy production is maximized. Under such conditions, aerosols
become the primary parameter influencing the distribution of solar irradiance among its
components. (e.g., Blaga et al., 2024). Specifically, the BRL model accounts for aerosols indirectly
through the clearness index, which is indicative of the overall atmospheric attenuation of solar
radiation.
In regions dominated by abundant sunshine, such as the Mediterranean and Middle East, which are
favorable for solar based power generation, the attenuation of solar irradiance is strongly influenced
by aerosols, and particularly desert dust aerosols. Several studies highlighted the impact of desert
dust aerosol in the downwelling solar irradiance and the energy production in these regions
(Fountoulakis et al., 2021; Papachristopoulou et al., 2022; Kosmopoulos et al., 2018; Kouklaki et al.,
2023). The significance of considering the effect of aerosols in short-term solar irradiance forecasting
and nowcasting is emphasized by Kazantzidis et al. (2017), Raptis et al. (2023) and
Papachristopoulou et al. (2024).
The Global Solar Energy Estimator (GSEE; Pfenninger & Staffell, 2016) is a widely used open access
model for simulating PV power output, designed for rapid calculations and ease of use. It comes with
an implementation of the BRL diffuse fraction model (Ridley et al., 2010; Lauret et al., 2013).
While PV power modelling is essential for linking solar resources to energy production, the existing
literature does not adequately address its reliability under diverse atmospheric conditions. To the
best of our knowledge, the existing literature does not include studies that explicitly address the
uncertainties in PV energy production modeling associated with the partitioning of solar radiation
into its direct and diffuse components at the model input. In this study, we supply GSEE with input
data from ground-based measurements as well as from the Copernicus Atmospheric Monitoring
Service (CAMS), aiming to investigate differences in PV power output simulations, which arise from
providing only GHI as input radiation data. At the outset, we focus on evaluating the reliability of BRL
under diverse atmospheric conditions, with particular attention to the dependence of its accuracy
on the presence of clouds and aerosols. To further explore this, we conduct a sensitivity analysis
using radiative transfer model (RTM) simulations under cloud-free skies. Following these analyses,
we assess the extent to which the associated uncertainties in the estimation of the diffuse fraction
spread to the power generation over hourly intervals. This step involves simulating PV plants with
varying configurations. GSEE is also effective for analyzing trends and variability in solar based power
generation through its climate interface submodule (e.g., Hou et al., 2021), where the BRL model is
integrated within the internal processing chain The accuracy of the climate interface in estimating
the total daily PV power output is also evaluated in this study.

**2. Data and Methodology**
2.1 Global Solar Energy Estimator (GSEE)
The modelling of the PV power output is conducted using the version 0.3.1 of GSEE (Pfenninger &
Staffell, 2016). The model features functions for simulating a complete PV system, incorporating
characteristics and specifications such as location, installed capacity, technology, tracking (fixed, 1-
axis, 2-axis), tilt angle, and orientation.
The user provides as input time-series data of solar radiation, and optionally, ambient air
temperature and surface albedo. Specifically, the model requires GHI and, when available, the
Diffuse Fraction. If the diffuse component is not provided, the provided implementation of the BRL
diffuse fraction model (Ridley et al., 2010; Lauret et al., 2013) is employed to estimate it, relying only
on time-series of the hourly clearness index and the geographical coordinates. While in the single-
site application of the GSEE model with hourly time resolution the user has the option to adjust the
input and select alternative diffuse fraction models implemented by external libraries, e.g., pvlib
(Anderson et al., 2013), the climate data interface automatically invokes the BRL model as part of the
internal processing workflow. GSEE utilizes the provided information for the distribution of the
irradiance components and applies trigonometric calculations to determine the total solar
irradiance incident on the panel's inclined plane. More precisely, for the plane-of -array irradiance
calculation a GSEE includes the submodule "trigon" (transposition model), which is based on
trigonometric formulations, that account of the surface albedo, thereby including the ground-
reflected component of solar radiation. However, the transposition model is integrated within the
GSEE internal algorithms, so it cannot be modified by the user.
After solar irradiance the most significant parameter regarding energy production is air temperature
(e.g., Dubey et al., 2013). If temperature is not provided by the user, the model assumes a default
value of 20 °C. In this study, temperature was used as input only in the simulations with BSRN data,
as it is provided alongside radiation measurements. A surface albedo value of 0.3 considered by
default from the model, introduces some uncertainty in our simulations, which however is estimated
to be small. Under cloudless conditions, a 10% difference in surface albedo changes the GHI by ~1%
for SZA < 75°. Differences are larger under cloudy conditions (~ 10% difference in GHI for a 10%
difference in surface albedo). Nevertheless, surface albedo at the selected sites is generally low and
relatively invariant throughout the year (even at the most northern site of Lindenberg there is only a
limited number of days with increased surface albedo due to snow cover).
The available options for the panel type are crystalline silicon (c-Si) and Cadmium Telluride (CdTe),
where the power output is modeled based on the relative PV performance model described by Huld
et al. (2010). For fixed panels, a built-in latitude dependent function for the optimal tilt is also
included.
Moreover, GSEE includes a Climate Data Interface submodule that enables the processing of gridded
climate datasets, with varying temporal resolutions, ranging from hourly to annual. Within the
context of this submodule, the use of BRL serves as part of the resampling and upsampling
processes applied to input climate datasets with daily resolution. For processing data with lower-
than-daily resolutions, it incorporates the use of Probability Density Functions (PDFs), which
describe the probability with which a day with a certain amount of radiation occurs within a month
(Renewables Ninja, n.d.). This methodology accounts for the non-linear distribution of mean monthly
radiation across individual days, ensuring a more representative temporal disaggregation. The
processes applied to the mean daily irradiance are described in detail in Section 3.4.
For the purposes of this study, we simulated solar plants with capacity of 1 kWp, and for both
available technologies. The simulations with c-Si technology, considered as default by the model,
are presented in detail the following sections. The results of the simulations with CdTe technology
are provided in the supplement, and are not thoroughly discussed, since they are very similar to the
results for the c-Si technology. Regarding the mounting approach, the solar plants were either static
and oriented to the south or equipped with a 2-axis solar tracking system. In the case of fixed panels,
we selected the optimal tilt angle relying on the latitude dependent built-in function.
The input parameters defining the characteristics of the simulated PV plants are summarized in Table
166 1.

**Table 1.** Input parameters defining the characteristics of the simulated PV plants

| Capacity | Mounting Approach | | 2-axis tracking | Technology | |
|----------|-------------------|---|-----------------|------------|------|
| 1 kWp | Fixed | | 2-axis tracking | c-Si | CdTe |
| | Orientation: south | Tilt Angle: f(latitude) built-in function for optimal tilt | | | |


2.2 Ground-based measurements
We supplied GSEE with ground-based irradiance as well as ambient temperature measurements
collected from five stations of the Baseline Surface Radiation Network (BSRN; Driemel et al., 2018).
Moreover, information about aerosols was retrieved from co-located stations of the Aerosol Robotic
Network (AERONET; Holben et al., 1998; Dubovik et al., 2000).
Information for the stations utilized for this study is summarized in Table 2, and their geographical
location is depicted in Figure 1.

**Table 2.** Detailed information about the location of the ground-based stations used in this study.

| STATION | Latitude [° N] | Longitude [° E] | Elevation [m] |
|---|---|---|---|
| Carpentras (CAR) | 44.08 | 5.06 | 100 |
| Cener (CNR) | 42.82 | -1.60 | 471 |
| Izaña (IZA) | 28.31 | -16.50 | 2373 |
| Lindenberg (LIN) | 52.21 | 14.12 | 125 |
| Tamanrasset (TAM) | 22.79 | 5.53 | 1385 |


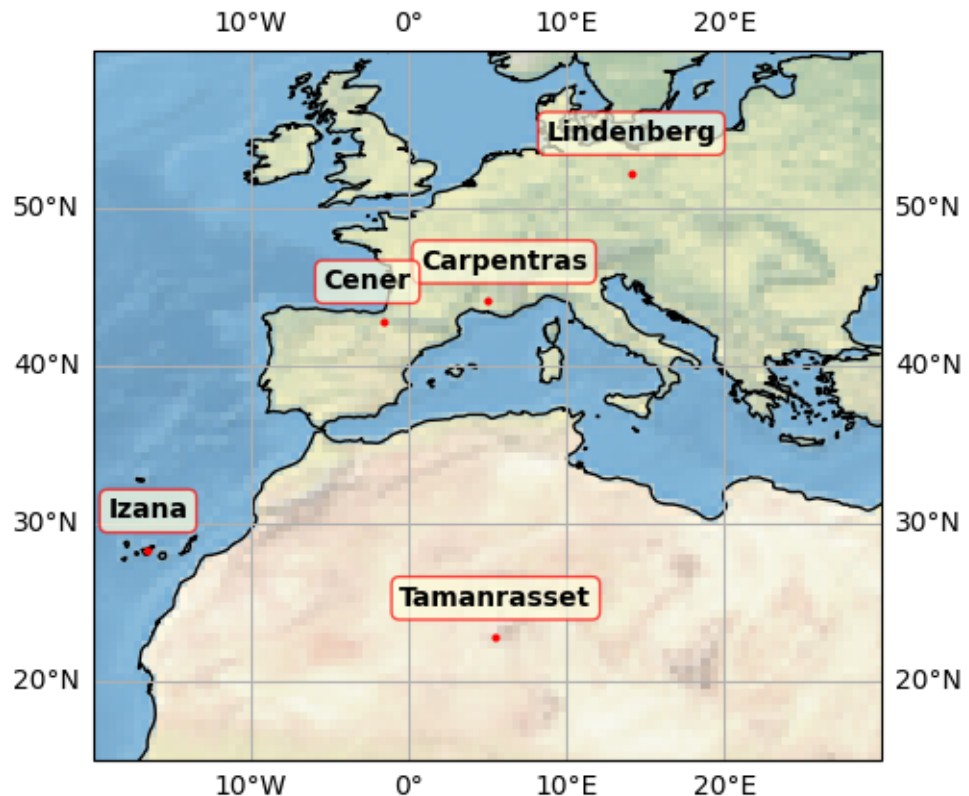


**Figure 1.** Locations of the BSRN and co-located AERONET stations that are used in the current

181                                                                    study


BSRN station-to-archive files were accessed and manipulated using the SolarData v1.1 R package
(Yang, 2019), and the BSRN-recommended quality check (QC) tests (Long & Dutton, 2010) applied
to the collected data. Some data gaps arose due to measurements removed during the QC
procedure. Although these data gaps are, in most cases, shorter than 2-3 hours, they may affect the
BRL performance throughout the corresponding days. Consequently, days affected by such data
gaps excluded from the analysis. We retrieved data for 2017, with 1-minute temporal resolution. We
used GHI, DHI, and Temperature as inputs to the GSEE model. Initially, the data were resampled to
hourly and mean hourly values of GHI and DHI are calculated. Then, the simulations were conducted
using either GHI and DHI, or only GHI along with the deployment of BRL. The input to BRL consists of
hourly clearness index, derived by dividing GHI measurements with the solar radiation incident on a
horizontal plane at the Top of the Atmosphere (TOA) above the examined location. Subsequently, the
1-min timeseries resampled also to a daily resolution and transformed into three-dimensional
arrays, $GHI = f(time, lat, lon)$, where the spatial dimensions of each dataset corresponded to a
unique point defined by the coordinates of the associated station. Simulations with the daily time-
resolved dataset were performed using the Climate Data Interface.
Representing cloudiness is a challenging task that requires several observations. For this purpose,
aiming to obtain an indicative measure of the intra-hour cloudiness conditions we adopted the
following formulation.  Specifically, measurements of Direct Normal Irradiance (DNI) were utilized to
obtain information for cloudiness relying on the conditions stated by WMO (2021), according to
which sunshine duration is the total period where DNI exceeds 120 $W/m^2$. Alternative approaches
such as the Cloud Modification Factor, require estimates of the clear sky irradiance, which
introduces additional uncertainty.  For the purpose of this analysis, we introduced a solar visibility
(SV) parameter. Specifically, we assigned the value 0 when sun was obscured and the value 1 when
visible. Aiming to describe the mean intra-hour cloudiness conditions, we considered the sky as
cloud-free, cloudy, and partly cloudy based on the mean SV for the entire corresponding hour as
follows:
$$\langle SV \rangle_{hour} : \begin{cases} 1 & cloud-free \\ \in (0,1) & partly\ cloudy \\ 0 & cloudy \end{cases}$$
For aerosol information, we accessed the AERONET Version 3 (V3) (Giles et al., 2019) and retrieved
level 2.0 data (from direct sun measurements) for Aerosol Optical Depth at 500nm ($AOD_{500}$), which
serves as a representative measure of the aerosol load; Ångström Exponent between 440 and 870
nm wavelengths ($AE_{440-870}$), where values near 0 correspond to coarse dust particles and values
around 2 to fine (e.g., smoke) particles (Dubovik et al., 2002); and Fine Mode Fraction at 500nm
($FMF_{500}$) obtained from the Spectral Deconvolution Algorithm (SDA) retrievals, to distinguish aerosol
into fine and coarse mode. The data were resampled at hourly intervals and a mean hourly value
calculated. After, the hourly mean values divided into clusters based on $AOD_{500}$, reflecting different
levels of aerosol load and allowing us to quantify their impact on solar energy production. To
investigate the impact related exclusively to aerosols, we included only hours with cloud-free sky
conditions. The clusters are defined in detail as follows:
•  $AOD_{500} \leq 0.05$: Low aerosol load
•  $0.05 < AOD_{500} \leq 0.15$: Moderate aerosol load
•  $0.15 < AOD_{500} \leq 0.3$: High aerosol load
•  $AOD_{500} > 0.3$: Very high aerosol load
To evaluate the performance of the Climate Interface over daily intervals, we defined the sunny
(cloudless) days using the condition: $\langle SV \rangle_{day} \geq 0.9$. Next, to characterize the average aerosol
conditions on sunny days, we applied the following classification:
•  $\langle AOD_{500} \rangle_{day} \leq 0.05$: very-low aerosol
•  $\langle AOD_{500} \rangle_{day} > 0.05$: aerosol-laden
Detailed comparisons of the energy production over hourly and daily integrals under the various
predefined sky conditions are provided in the supplement through evaluation metrics.
The selected locations have quite different atmospheric conditions regarding cloudiness and
aerosols. Additionally, they vary in altitude. A brief overview of the prevailing conditions derived from
the ground-based data is provided on the supplement. Regarding cloudiness, it is notable that in
Lindenberg the sky is generally overcast, whereas in southern locations sunshine dominates. In
terms of aerosols, very high aerosol loads occur more frequently in Tamanrasset. As for aerosol type,
there is considerable variation among the examined locations: Carpentras, Cener, and Lindenberg
are primarily influenced by fine mode aerosols, while Tamanrasset and Izaña are mostly affected by
coarse mode aerosols.
For investigating the impact of desert dust aerosol in solar based power generation, Tamanrasset
serves as a representative and exceptional case because it is in a region with important sources of
Saharan dust aerosols (Faid et al., 2012). Meanwhile, Izaña, located in subtropical North Atlantic, is
a high mountain station within the free troposphere, affected mineral dust when the Saharan Air
Layer top exceeds the station height, especially through August to October (Toledano et al., 2018;
Cuevas et al., 2018). Due to its high altitude, Izaña avoids contamination from local or regional
sources (Barreto et al. 2022). The Canary Islands, where Izaña is located, are influenced by extreme
dust events that cause a significant decrease in PV power generation (Canadillas-Ramallo et al.,
2021). In South Europe, which is also affected by the transport of Saharan dust across the
Mediterranean, aerosol types exhibit a mixture as a result of simultaneous local pollution and low
concentration of mineral dust (Logothetis et al., 2020).
2.3 Copernicus Atmospheric Monitoring Service (CAMS)
We retrieved data from the CAMS radiation service (Schroedter-Homscheidt et al., 2022; Qu et al.,
2017), from the solar radiation time-series product (CAMS, 2020). The CAMS solar radiation service
provides historical estimates for global solar radiation, along with its components, from 2004 to
present. These values are provided with a frequency as fine as 1-minute. In this study, we used the
hourly time-series of GHI and DHI for all-sky conditions, setting the input coordinates to match the
locations of the BSRN stations. The solar radiation time-series product (CAMS, 2020) performs
interpolations integrated in its internal algorithm and provides time-series for the coordinates and
the altitude of a single-site location. We compared the solar energy production derived from the use
of CAMS data with that derived from the use of ground-based measurements from BSRN.
2.4 Radiative Transfer Model (RTM)
We performed Radiative Transfer (RT) simulations aiming to further assess the uncertainties in
estimating the diffuse fraction arising from the effect of aerosols. The simulations were conducted
using libRadtran (Emde et al., 2016; Mayer & Kylling, 2005), a widely used software package, allowing
the computation of radiances, irradiances, and actinic fluxes. A sensitivity analysis was performed
by comparing the diffuse irradiance calculated from libRadtran with the estimations of BRL. This
analysis examines the dependence of the aerosol-related discrepancy as function of Solar Zenith
Angle (SZA) and latitude, considering the effect of parameters such as surface albedo and altitude.
To conduct aerosol parameterizations, we considered the default aerosol extinction profile (Shettle,
1989) and set asymmetry factor (gg) to 0.7, while varying the Single Scattering Albedo (SSA) and the
Ångström Exponent (AE), and defining $AOD_{500}$ by adjusting the value of the parameter-b in
Ångström's law (Ångström, 1929) as follows:
$$\tau_\lambda = b \cdot \lambda^{-a} \rightarrow AOD_{500} = b \cdot (0.5 \, \mu m)^{-AE}$$
The standard aerosol profiles (Anderson et al., 1986) were used for all sites. According to
Fountoulakis et al. (2022), using a more accurate vertical distribution of aerosols in the troposphere
would have a negligible effect in the GHI and DHI at the Earth's surface.
Table 3 illustrates the libRadtran settings used in this study.
**Table 3.** LibRadtran inputs

| Parameter | Input |
|---|---|
| Atmospheric profile | Mid-latitude summer (April-September)/mid-latitude winter (October - March) (Anderson et al., 1986) |
| Extraterrestrial spectrum | (Kato et al. 1999) |
| Datetime | date and time input accompanied by project location coordinates |
| Altitude | 0.1/2 km |
| Surface albedo | 0.2 / 0.8 |
| Number of streams | 6 |
| RT solver | sdisort (Buras et al., 2011) |
| AE | 0 – 2 with step 1 |
| SSA | 0.7, 0.9, 1.0 |
| gg | 0.7 |
| TOC (Total Ozone Column) | 300 DU |
| Integrated Water Vapor | 15 mm |


## 3. Results

### 3.1 Performance verification of the BRL diffuse fraction model

The performance of BRL was evaluated by comparing the actual diffuse fraction, obtained directly from resampled to hourly BSRN ground-based measurements, with that derived using BRL. As a first step, to isolate the influence of SZA from that associated with the atmospheric conditions, the difference in diffuse fraction (DF) between the observed and the one estimated using BRL as a function of SZA is presented in Figure 2. The atmospheric conditions are represented separately for both all-sky and cloud-free sky conditions and are grouped into clusters, as outlined in Section 2.2. The patterns reflecting the differences under the distinct sky conditions indicate an additional dependency on SZA, which becomes apparent approximately at SZA between 60° and 70°. In most cases, there is an almost constant displacement with respect to y=0 below 60°, as well as a change in behavior when SZA exceeds this value. Izaña presents a special case, as the station is located at a very high altitude. At such high altitudes the contribution of the diffuse component to the total irradiance is significantly smaller relative to lower altitude sites, which seems to be captured more accurately by BRL at high SZAs. We must also note that (i) at Izaña, the actual diffuse irradiance may experience an additional enhancement due to the contribution of adjacent lower-lying clouds – an effect that is not accounted for in the diffuse fraction model, and (ii) during dust events the site is usually inside – and not under – the dust layer, which results in more complex interactions between dust and solar radiation relative to lower altitude sites. Defining an exact limit (for the lower altitude sites), where the behavior is changing, is challenging; therefore, 60° was selected for practical energy-related applications, focusing on periods with meaningful energy contribution, and is supported by the sensitivity analysis (Section 3.2) under clear-sky conditions. Concerning the same grouped atmospheric conditions, Figure 3 illustrates the comparison between the observed and the estimated diffuse fraction for $SZA \leq 60°$. This approach allows us to examine BRL performance after eliminating the influence of SZA, thereby providing a more comprehensive view of its reliability.

$$\text{Diffuse Fraction (DF)}_{observed} - \text{Diffuse Fraction (DF)}_{BRL} = \mathbf{f}(\text{sza})$$

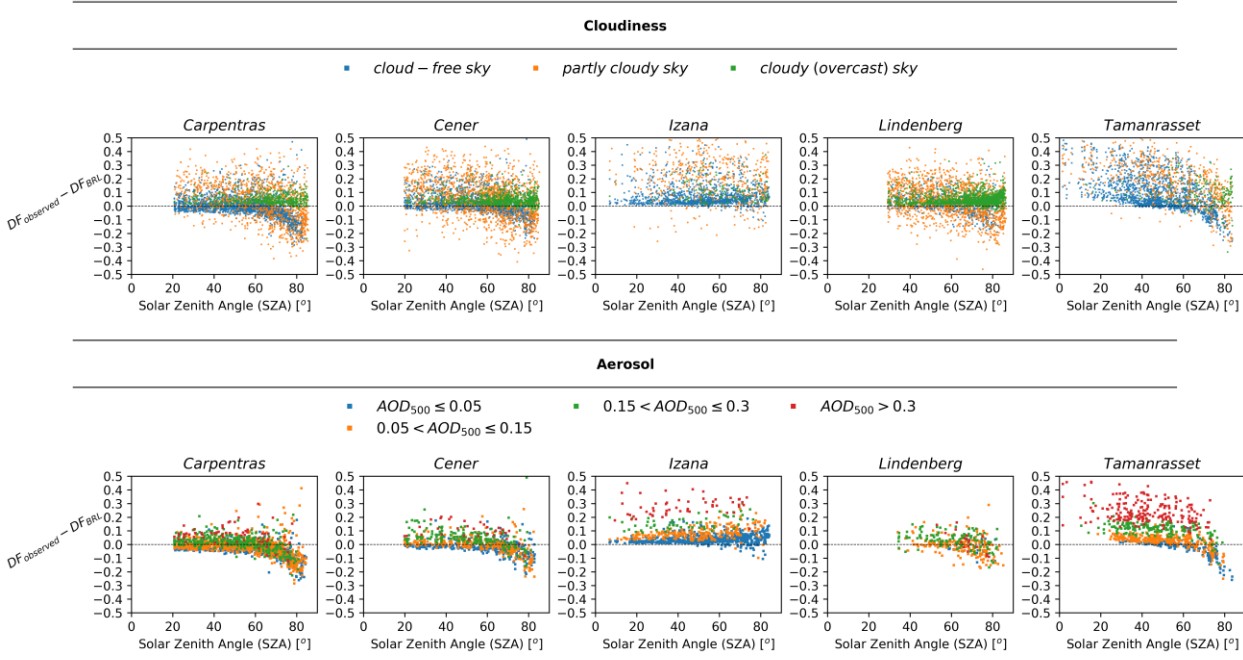

**Figure 2.** Difference between the diffuse fraction estimated by the ground-based measurements and by using the BRL model as a function of SZA under diverse atmospheric conditions: (top) classification with respect to cloudiness and (bottom) classification with respect to aerosol optical depth

$$\text{Diffuse Fraction (DF)}_{BRL} = \mathbf{f}(\text{Diffuse Fraction (DF)}_{observed})$$

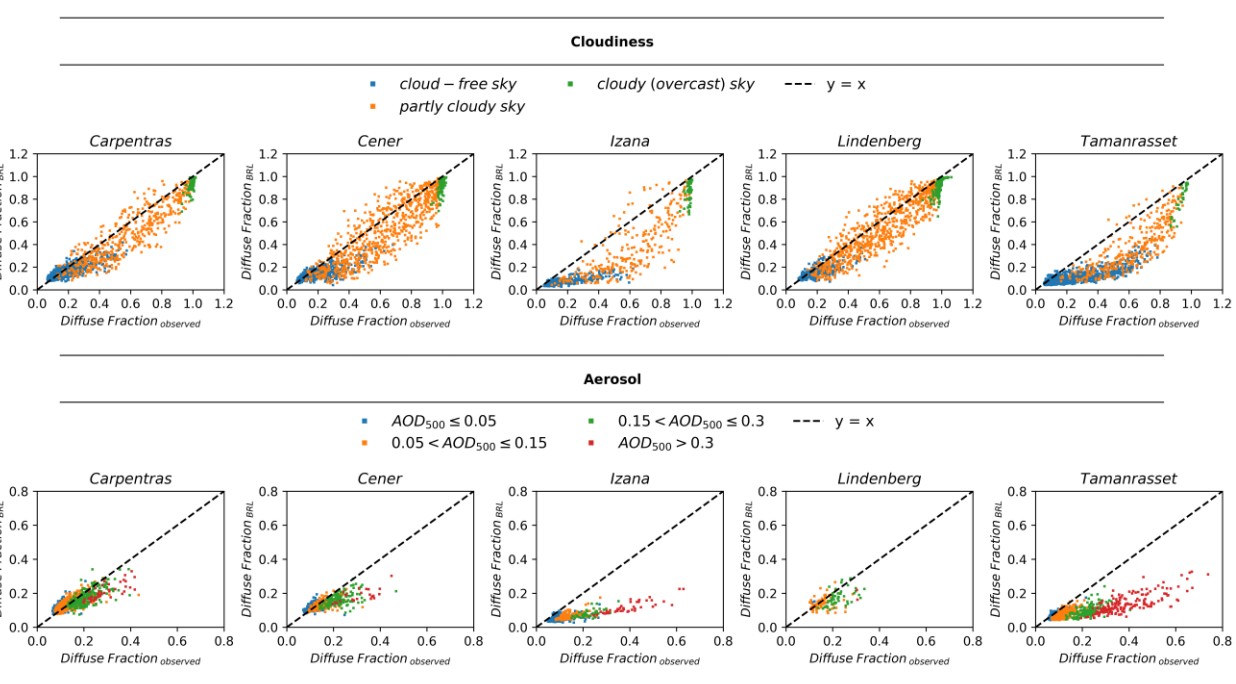

**Figure 3.** Comparison of the diffuse fraction estimated using BRL with that estimated by the
ground-based measurements under diverse atmospheric conditions for SZA < 60°: (top)
classification with respect to cloudiness and (bottom) classification with respect to aerosol optical
depth

From Figure 3, a distinct dependency of BRL's reliability on the atmospheric conditions can be
observed. Under all-sky conditions, the presence of clouds has a notable impact on the model's
performance. Partly cloudy conditions result in greater dispersion of the values from the identity line
respectively, likely due to the complexity of such sky scenes. Under overcast conditions, where the
sky can be considered homogeneous and isotropic, the model in most cases performs slightly better.
However, the limitations of the DNI-based classification methodology, related to the complexity of
the cloud scenes, the spatiotemporal variability during the hourly periods, and the 3D variability of
cloud properties, would require additional observational tools for a more detailed investigation. More
specifically, the vast majority of overcast cases where the BRL diffuse fraction is below 0.8 while the
observed is close to 1 correspond to periods involving rapid transitions between partly cloudy and
overcast skies, occurring either during the hour itself or immediately before or after it. Furthermore,
a limited number of cases identified during intense dust events at Tamanrasset and Izana, where the
reduction of DNI was so pronounced that the applied DNI-based criterion classified these conditions
as overcast. However, these cases are not further investigated, as the energy production levels during
such periods are very low.
Under cloud-free skies, BRL tends to underestimate, and this bias becomes more pronounced as
aerosol load increases. Aiming to highlight this dependency, Figure 4 shows the difference between
the estimated and the observed diffuse fraction as function of $AOD_{500}$, emphasizing also the extent
to which it is related to the aerosol type by providing $FMF_{500}$. A decrease for increasing $AOD_{500}$ is
evident across all cases. In Tamanrasset and Izaña, associated with the influence of Saharan dust,
the coarse mode dominates, and a more distinct and well-defined curve is depicted compared to
other sites.
It is important to clarify that for assessing the impact of aerosols we have assumed entirely cloud-
free conditions. However, the criterion applied based on DNI does not fully guarantee the absence of
small, scattered clouds within the sky dome. Such clouds could induce slight enhancements in DHI.
A more rigorous assessment of the impact associated exclusively with aerosols could be achieved
by integrating images from ground-based co-located all-sky cameras. On the other hand, the
presence of aerosols even under cloudy scenes, introduces an additional uncertainty which is
difficult to investigate accurately.

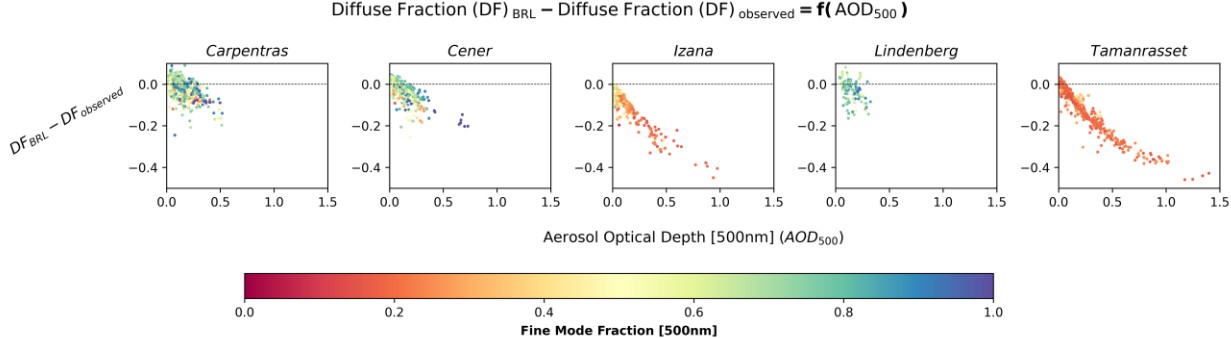


**Figure 4.** Difference between the estimated using BRL and thediffuse fraction estimated by the
ground-based measurements as function of $AOD_{500}$ and $FMF_{500}$

3.2 Sensitivity analysis of the BRL performance under cloud-free sky conditions from RT
simulations
The uncertainties in estimating diffuse fraction under cloud-free sky conditions, as discussed in
section 3.1, are further investigated. We performed RT simulations using libRadtran to calculate GHI
and DHI under various aerosol scenarios. The resulting GHI values were then used as input to BRL to
estimate the diffuse fraction, which was subsequently compared to the diffuse fraction derived
directly from the ratio of DHI to GHI computed by libRadtran.
To ensure a comprehensive analysis, we considered three representative latitudes (25°, 35° and 45°).
Since BRL requires an hourly time-series of GHI as input, the analysis was conducted for the summer
solstice. On this day, a sufficient number of hourly values are available, corresponding to a wide
range of SZA values, allowing for a robust assessment of the methodology. The sensitivity analysis
was performed for surface albedo values of 0.2 and 0.8 as well as for altitudes of 0.1 and 2 km. For
aerosol parameterization, we examined completely clear-sky conditions as a reference, alongside
scenarios with $AOD_{500}$ values of 0.2, 0.6, and 1, while varying the SSA and AE. Specifically, the
scenarios included SSA values of 0.7, 0.9 and 1, combined with AE values of 0, 1 and 2. The results
of this sensitivity analysis for an albedo of 0.2 are provided in Figure 5, while the results for an albedo
of 0.8 are included in the supplement (Figure S1).
The results confirm that BRL performs well under clear sky conditions and for SZA below 60°, while
the incorporation of aerosols in the sky scene introduces larger uncertainties. In all scenarios, we
observe that lower values of AE correspond to higher uncertainties. Moreover, when SSA is 0.9 or 1
BRL gradually tends to underestimate the diffuse fraction as aerosol load increases. Instead, when
SSA is 0.7, BRL exhibits a different behavior, shifting toward an overestimation of the diffuse fraction
at high aerosol loads.
The findings of this sensitivity analysis are consistent with the evaluated BRL performance from
ground-based measurements presented in section 3.1, especially at SZA smaller than 60° - 70°, and
underscore the role of aerosol in the accuracy of diffuse fraction estimations. Differences between
the results shown in Figures 2 and 5 at SZA between 60° - 80° can be due to a number of site-related
reasons. For example, enhancement of the diffuse component due to scattering by underlying
atmospheric layers and clouds in the case of Izaña may compensate the observed overestimation of
the diffuse fraction by BRL. Concerning the impact related to AE and SSA, we confirm that the higher
underestimations observed for Tamanrasset and Izaña are associated with the optical properties of
desert dust aerosol particles. While AE and SSA alone are not sufficient to fully characterize the
aerosol type, they serve as strong indicators, aligning with the classification framework of Dubovik et
al. (2002). The same comparison for albedo 0.8 (Figure S1 in the supplement) reveals a significant
broadening of the discrepancies. Moreover, we observe the presence of a systematic error, even
under clear sky conditions.
The resulting differences were practically identical across the three selected latitudes, indicating
that the BRL model is largely independent of latitude and can therefore be considered as a reliable
solution over a wide range of latitudes. Furthermore, the effect of altitude was found to be small.
Finally, the outcomes of this analysis highlight potential inconsistencies arising from aerosols with
different optical properties. Although the updated parameters of the BRL's model (as implemented
in the GSEE model) reported by Lauret et al. (2013) were derived using data from nine worldwide
locations, encompassing a broad range of sky conditions that capture a fully representative set of
optical properties remain challenging.

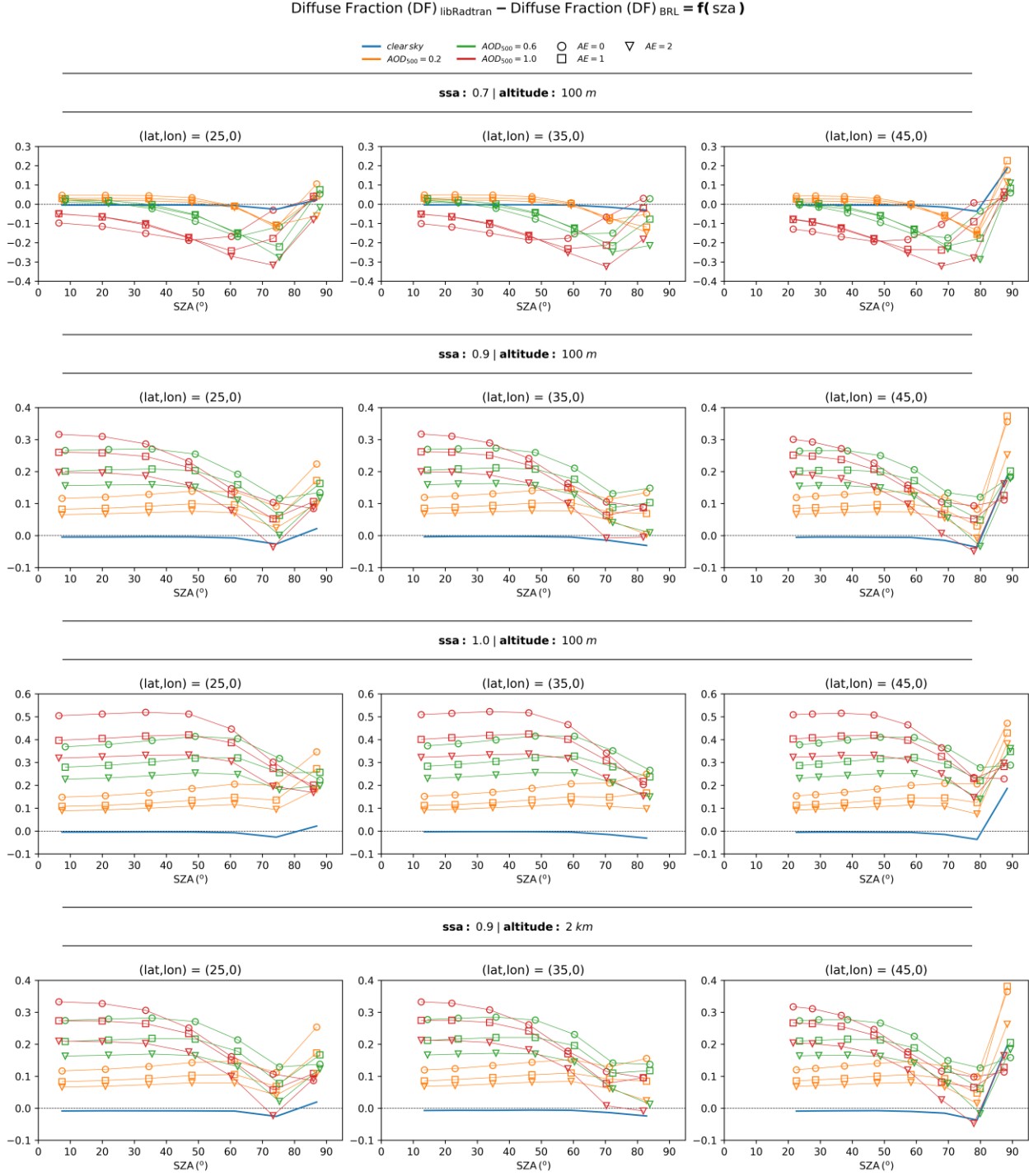

**Figure 5.** Difference between the diffuse fraction derived directly from the computations of DHI and GHI using libRadtran and the one estimated by applying BRL to the libRadtran-computed GHI

## 3.3 Analysis of the differences in energy production using hourly integrals within the modelling of PV
plants
Uncertainties in estimating the diffuse fraction influence the calculation of the total irradiance
received by an inclined panel's surface, thereby affecting the accuracy of the PV power simulations.
In this section, we employ the main submodule of GSEE, used for modelling the electric output from
a PV panel, aiming to assess the extent to which these uncertainties propagate to the estimation of
the hourly power production. We analyze discrepancies arising from using only GHI from BSRN as
input radiation data to the model, instead of both DHI and GHI. More specifically, we compare the
total energy produced per hour per unit, expressed in watt-hours (Wh), per unit of nominal power
(kWp). The energy production is evaluated for both fixed panels and 2-axis tracking systems.
The results of this comparison for c-Si based technology PV panels for different atmospheric
conditions are presented in Figure 6, illustrating the impact of cloudiness, and in Figure 7,
demonstrating the effect of aerosols. The corresponding results for CdTe technology are provided in
the supplement (Figures S2 and S3 respectively). In the modelling of 2-axis solar tracking systems,
where the panel is continuously adjusted to maintain a perpendicular orientation to incoming solar
radiation, the system becomes more sensitive to uncertainties in the estimation of the diffuse
fraction, leading to more significant differences in energy production. Specifically, the contribution
of the direct irradiance is maximized in such systems, as the panel exploits the entirety of the
available direct irradiance. On the other hand, in the simulation of static panels, the contributions of
direct and diffuse components are more evenly distributed, making the impact of diffuse fraction
uncertainties less pronounced in energy production.
Regarding the uncertainties related to the atmospheric conditions, from Figure 6 we confirm that the
highest dispersion occurs in partly cloudy conditions, while from Figure 7, where we examine cloud-
free conditions, we note that further improvement achieved as aerosol load decreases. Under totally
overcast skies the energy production is extremely low, rendering errors practically negligible.
Moreover, accuracy is influenced by aerosols, where a gradual decline in accuracy is detected as
aerosol load increases. However, assessing the extent of aerosol loading impact is complex,
depending on the interaction of solar radiation with particles of varying optical properties, as
extensively analyzed in the previous sections. This effect becomes particularly evident in cases of
high aerosol loading, where a noticeable offset is observed, while under certain conditions, the
associated uncertainty is comparable to that found in partly cloudy conditions.

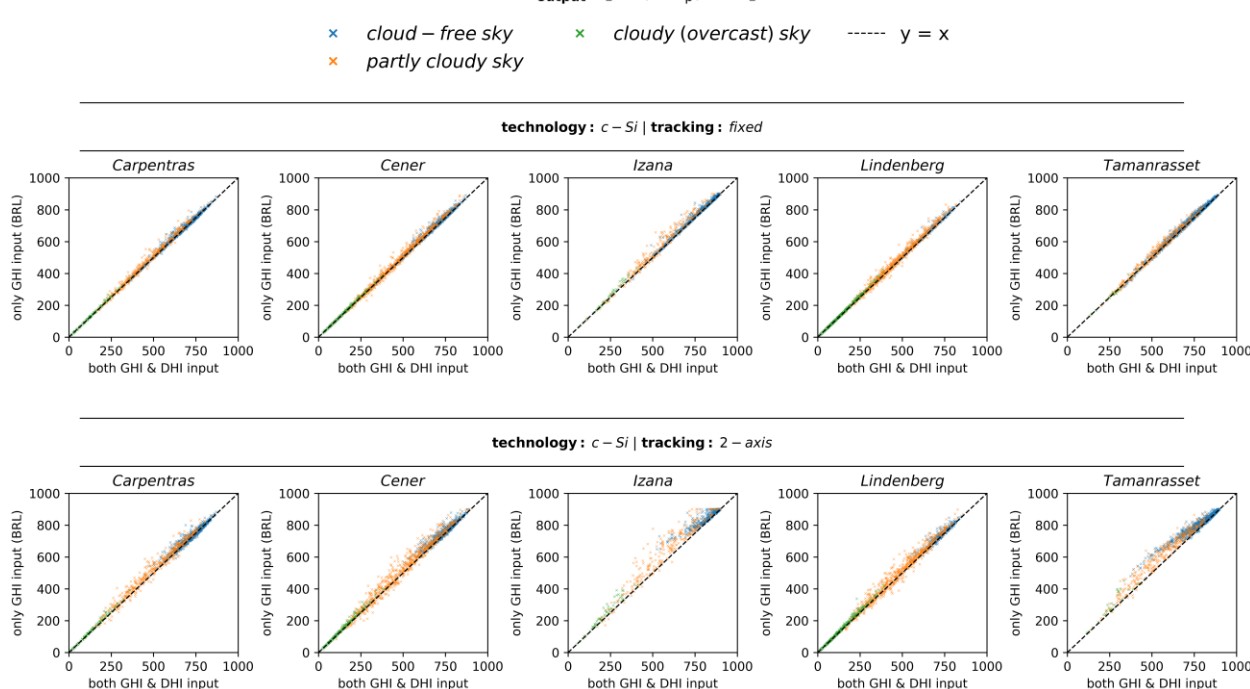


**Figure 6.** Comparison of the estimated hourly PV power generation between simulations performed

431         using GSEE with input data consisting of either only GHI or both GHI and DHI under varying

432                 cloudiness conditions: (top) fixed panels (bottom) 2-axis tracking systems

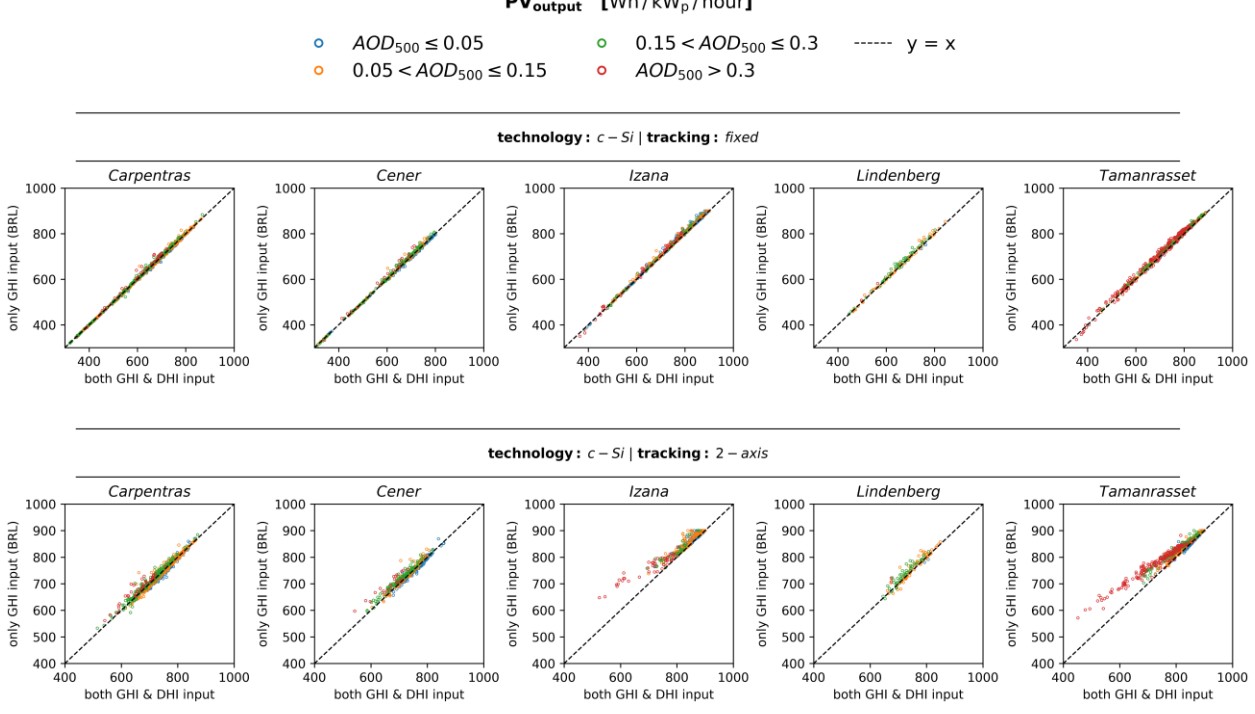


 **Figure 7.** Comparison of the estimated hourly PV power generation between simulations performed

using GSEE with input data consisting of either only GHI or both GHI and DHI under varying aerosol

436            conditions: (top) fixed panels (bottom) 2-axis tracking systems

The PV systems considered in this study have a nominal capacity of 1 kWp. The PV model applies a
default system loss factor of 10%. This effectively limits the maximum achievable power output to
approximately 90% of the nominal capacity (i.e., around 900 W/kWp). This effect becomes apparent
at the Izaña site due to its low latitude combined with its specific geographical and atmospheric
conditions, which lead to high irradiance levels. As a result, the simulated PV output in some cases
appears capped around 900 Wh/kWp per hour when only GHI is used.
Additionally, Tables 4 and 5 present the validation results for Carpentras and Tamanrasset, selected
as representative locations that encompass a wide variety of sky conditions. Validation results for
the remaining stations are available in the supplement (Tables S1-S3). All the evaluation metrics
correspond to simulations of PV panels with c-Si technology.
**Table 4.** Evaluation metrics for GSEE performance within hourly intervals in Carpentras, comparing
simulations with diffuse fraction from measurements and from the BRL model

| STATION: Carpentras | | fixed panels | | | 2-axis tracking | | |
|---|---|---|---|---|---|---|---|
| | | RMSE (Wh/kWp/hour) | MAE (Wh/kWp/hour) | rMBE (%) | RMSE (Wh/kWp/hour) | MAE (Wh/kWp/hour) | rMBE (%) |
| All-Sky scenes | | 12.6 | 6.6 | 0.8 | 20.8 | 12.5 | 1.2 |
| All-Sky scenes (cloudiness) | cloud-free | 9.2 | 4.6 | 0.4 | 14.8 | 8.7 | 0.5 |
| | partly cloudy | 19.5 | 12.5 | 2.3 | 32.5 | 23.9 | 3.8 |
| | cloudy (overcast) | 5.8 | 3.0 | 2.0 | 10.5 | 6.1 | 4.6 |
| Cloudless-Sky scenes (aerosol load) | low | 4.7 | 3.4 | -0.4 | 9.5 | 7.5 | -0.8 |
| | moderate | 4.3 | 2.2 | 0.1 | 7.8 | 4.7 | 0.0 |
| | high | 6.4 | 4.0 | 0.6 | 11.0 | 7.8 | 0.9 |
| | very high | 14.9 | 10.2 | 1.6 | 22.7 | 17.2 | 2.6 |


**Table 5.** Evaluation metrics for GSEE performance within hourly intervals in Tamanrasset,
comparing simulations with diffuse fraction from measurements and from the BRL model.

| STATION: Tamanrasset | fixed panels | | | 2-axis tracking | | |
|---|---|---|---|---|---|---|
| | RMSE (Wh/kWp/hour) | MAE (Wh/kWp/hour) | rMBE (%) | RMSE (Wh/kWp/hour) | MAE (Wh/kWp/hour) | rMBE (%) |
| All-Sky scenes | 13.6 | 9.3 | 1.0 | 40.4 | 27.8 | 3.8 |

| | | | | | | | |
|---|---|---|---|---|---|---|---|
| All-Sky scenes (cloudiness) | cloud-free | 11.5 | 8.0 | 0.8 | 35.3 | 23.4 | 2.9 |
| | partly cloudy | 20.1 | 15.0 | 2.0 | 56.1 | 45.7 | 8.1 |
| | cloudy (overcast) | 8.4 | 5.2 | -0.1 | 45.3 | 30.1 | 11.2 |
| Cloudless-Sky scenes (aerosol load) | low | 3.2 | 2.0 | 0.2 | 6.6 | 4.0 | 0.3 |
| | moderate | 5.4 | 4.6 | 0.6 | 13.0 | 10.5 | 1.2 |
| | high | 12.5 | 11.7 | 1.6 | 30.1 | 27.4 | 3.4 |
| | very high | 18.0 | 16.2 | 1.9 | 57.0 | 49.2 | 6.8 |


Based on the calculated statistical indices, the Root Mean Square Error (RMSE) values for fixed
panels range from 4.7 Wh/kWp/hour (clear sky) to 19.5 Wh/kWp/hour (partly cloudy) in Carpentras,
and from 3.2 to 20.1 Wh/kWp/hour in Tamanrasset. Under very high aerosol loading, RMSE reaches
14.9 and 18.0 Wh/kWp/hour, respectively. For 2-axis tracking systems, RMSE values vary
significantly, ranging from 9.5 to 32.5 Wh/kWp/hour in Carpentras and from 6.6 to 56.1 Wh/kWp/hour
in Tamanrasset, with peaks of 22.7 and 57.0 Wh/kWp/hour under very high aerosol loading
conditions. Similarly, the Mean Absolut Error (MAE) values are generally lower for fixed panels (3.4-
12.5 Wh/kWp//hour in Carpentras, 2.0-15.0 in Tamanrasset) and substantially higher for 2-axis
tracking (7.5-23.9 and 4.0-45.7 Wh/kWp/hour, respectively). Notably in Tamanrasset, MAE values
under very high aerosol loading exceed those observed under partly cloudy conditions, with values
increasing from 15.0 to 16.2 Wh/kWp/hour for fixed panels and from 45.7 to 49.2 Wh/kWp/hour for
2-axis tracking systems. Regarding the relative mean bias (rMBE), this remains mostly within ± 4.6%
for fixed panels but can reach up to 11.2% for 2-axis tracking, particularly in aerosol-laden
conditions.

3.4 Estimating total daily PV power output using the Climate Interface
Validation of the estimated daily energy production using the Climate Interface is achieved by
comparing the estimates with the results obtained from the direct summation of the hourly
simulations with input both GHI and DHI.
The Climate Interface generates the hourly profile of GHI for each day as a sinusoidal function. Then,
the BRL is applied to the hourly time-series, and the hourly power generation is computed. Finally,
these values are summed up to provide an estimate of the total daily output power. As shown in Fig.
8, which illustrates the differences between the Climate Interface estimates and the sums of the
hourly simulations, this approach introduces a variability throughout the year. Furthermore, Figure
S6 in the supplement presents the percentage differences between the two approaches, using the
latter as the reference.

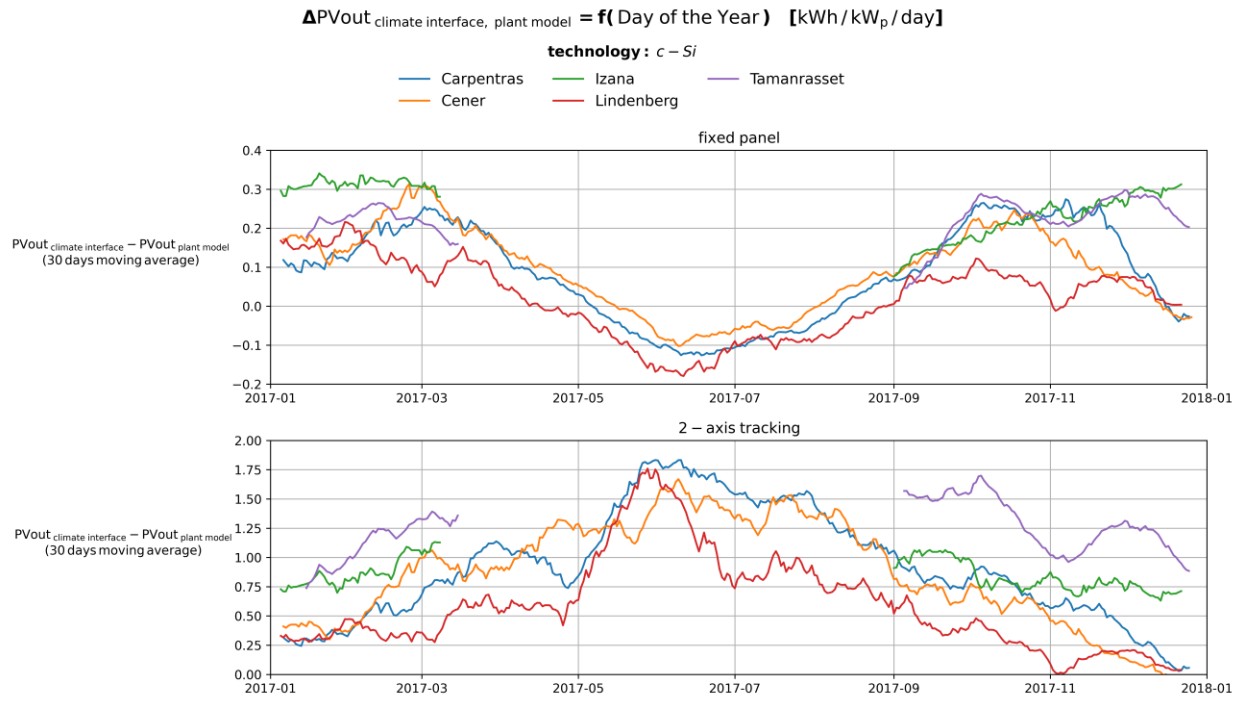


**Figure 8.** Time-series of the differences between the daily PV output estimated using the climate

481        interface and the corresponding daily sums from hourly simulations.


The time-series represent the centered 30-day moving average. To ensure that the values are
representative of the reference period, we have applied all conditions requiring at least 20 days of
available data within each 30-days interval. In Tamanrasset and Izaña, especially during the summer
months, there are significant data gaps on several days, often occurring around solar noon.
More precisely, from Fig. 8, we observe that within the modelling of PV plants with fixed panels, there
is a tendency to overestimate in winter, with deviations of approximately 0.3 kWh/kWp/day, and to
slightly underestimate in summer, where deviations are around 0.1 kWh/kWp/day. In contrast, for 2-
axis solar tracking systems, the resulting deviations are significantly larger, with a general tendency
toward overestimation that peaks during summer, reaching approximately 1.75 kWh/kWp/day. The
percentage differences span from -10 to 20 % for fixed panels and from -5 to 35 % for 2-axis tracking
systems.
The variability in the percentage difference between the daily PV output estimated using the climate
interface and the corresponding daily sums is mainly a function of the minimum SZA, while
especially in the case of modeling for 2-axits tracking systems, the variation is also influenced by
aerosol loading, with differences tending to increase as aerosol load rises (Figures S4 and S5 in the
supplement).
Additional validation results are provided in the supplement (Tables S4-S8). Indicatively, for
Carpentras and Tamanrasset, representative results are discussed below.  For fixed panels, RMSE is
minimized at 0.18 kWh/kWp/day under very-low aerosol conditions, compared to the overall 0.22
kWh/kWp/day for Carpentras. In Tamanrasset, the lowest RMSE is observed at 0.15 kWh/kWp/day
under very low aerosol conditions, while the overall reaches 0.24. In the case of 2-axis tracking, a
significant increase is observed from low-aerosol to aerosol-laden conditions, ranging from 0.82 to
1.28 kWh/kWp/day in Carpentras and from 0.66 to 1.37 in Tamanrasset. Similar widening trends are
also evident in the MAE values across different aerosol loading conditions. The computed statistical
indices confirm that the differences are minimized under sunny and nearly aerosol-free sky
conditions. Comparing the performance on low-aerosol days to that on aerosol-laden, we conclude
that, particularly in the case of modelling 2-axis tracking systems, errors increase significantly. In
Tamanrasset, in particular, the errors are more than double.
3.5 Evaluation of the reliability of using the CAMS solar radiation time-series product in modelling
PV power potential
The aim of this section is to inspect the reliability of using the CAMS solar radiation time-series
product in modelling the PV power potential adapted to a certain location. A review of the existing
literature indicates a lack of studies directly examining the accuracy of using CAMS data for
assessing PV power potential. This is addressed by comparing the output power obtained from using
CAMS solar radiation data with that calculated using ground-based measurements. The analysis
focuses on the capability of CAMS to provide accurate estimates of both GHI as well as its individual
components.
In this section, we have excluded Izaña, because, due to its high altitude – as indicated through a
personal communication with Yves-Marie Saint-Drenan (2025) – comparable results would require
adjusting the measurements to the elevation of the stations, which is a complicated process and
beyond the scope of this study.
The CAMS-based diffuse fraction, compared to the observed, is presented in Figure 9 under different
prevailing conditions. We observe that the calculation of the diffuse component is subject to
significant uncertainty. Cloudiness is the primary uncertainty source, particularly under partly cloudy
conditions. Additionally, notable discrepancies related to aerosols emerge only in cases of very high
aerosol loading.

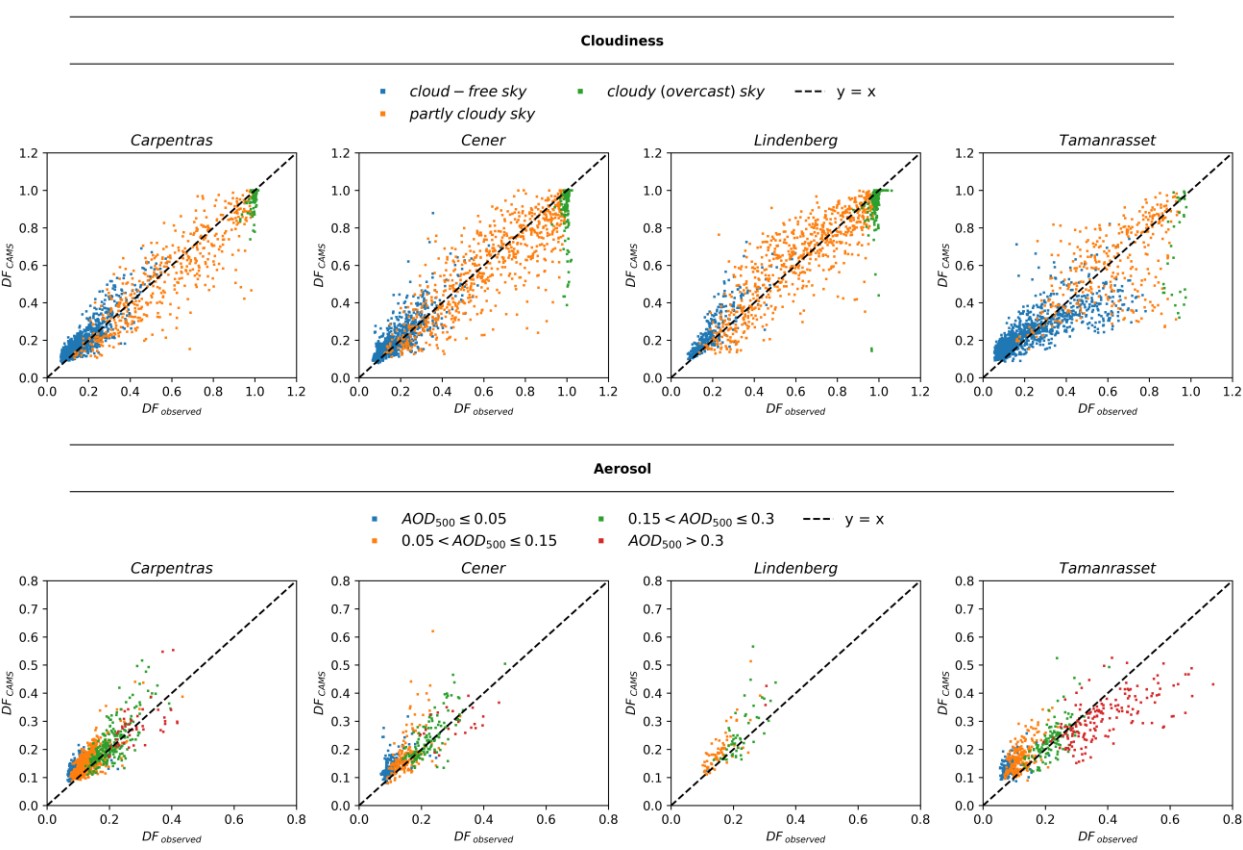

**Figure 9.** Comparison of the CAMS-based diffuse fraction estimated using BRL with the actual one
under diverse atmospheric conditions

In Fig. 10 we provide density scatter plots comparing the CAMS-based PV output power with that
computed from the ground-based BSRN data, aiming to illustrate how the uncertainty in the diffuse
component estimates propagate to the calculation of power generation. Notably, there is a much
greater dispersion from the y=x line in the case of simulating PV plants with 2-axis tracking system,
compared to that within the modelling of fixed panels. This outcome is attributed to the increased
sensitivity of the 2-axis tracking systems to the partitioning of global irradiance into its components.
Nevertheless, correlation coefficients are in all cases better than 0.9.
Additional evaluation metrics are provided in the supplement (Tables S9-S12). Indicatively, we
observe that under cloudless conditions, for fixed panels, RMSE ranges between 25.0 to 42.3
Wh/kWp/hour in Carpentras and 16.6 and 31.0 Wh/kWp/hour in Tamanrasset, with variations linked
to aerosol loading. Similarly, MAE ranges from 20.0 to 36.9 Wh/kWp/hour in Carpentras and 11.9 to
22.9 Wh/kWp/hour in Tamanrasset. For 2-axis systems, RMSE and MAE follow similar trend, ranging
from 28.8 to 49.9 Wh/kWp/hour and 22.3 to 44.1 Wh/kWp/hour, respectively, in Carpentras, and from
20.8 to 48.0 Wh/kWp/hour and 15.3 to 35.5 Wh/kWp/hour, respectively, in Tamanrasset. Conversely,
under cloudy conditions the errors are significantly increasing. In Carpentras, as well as in Cener,
and Lindenberg (according to the corresponding tables in the supplement) the errors peak under
partly cloudy conditions, with RMSE reaching up to 94.2 Wh/kWp/hour in Carpentras. However, in
Tamanrasset, the highest errors occur under overcast conditions, where RMSE and MAE for 2-axis
solar tracking systems reach 210.7 and 151.6 Wh/kWp/hour, respectively. This exception can be
interpreted through Figure 15, which illustrates that in the rare overcast scenes in Tamanrasset,
CAMS occasionally reports low diffuse fraction values instead of values close to 1, suggesting that
CAMS did not accurately represent cloudiness in these cases.

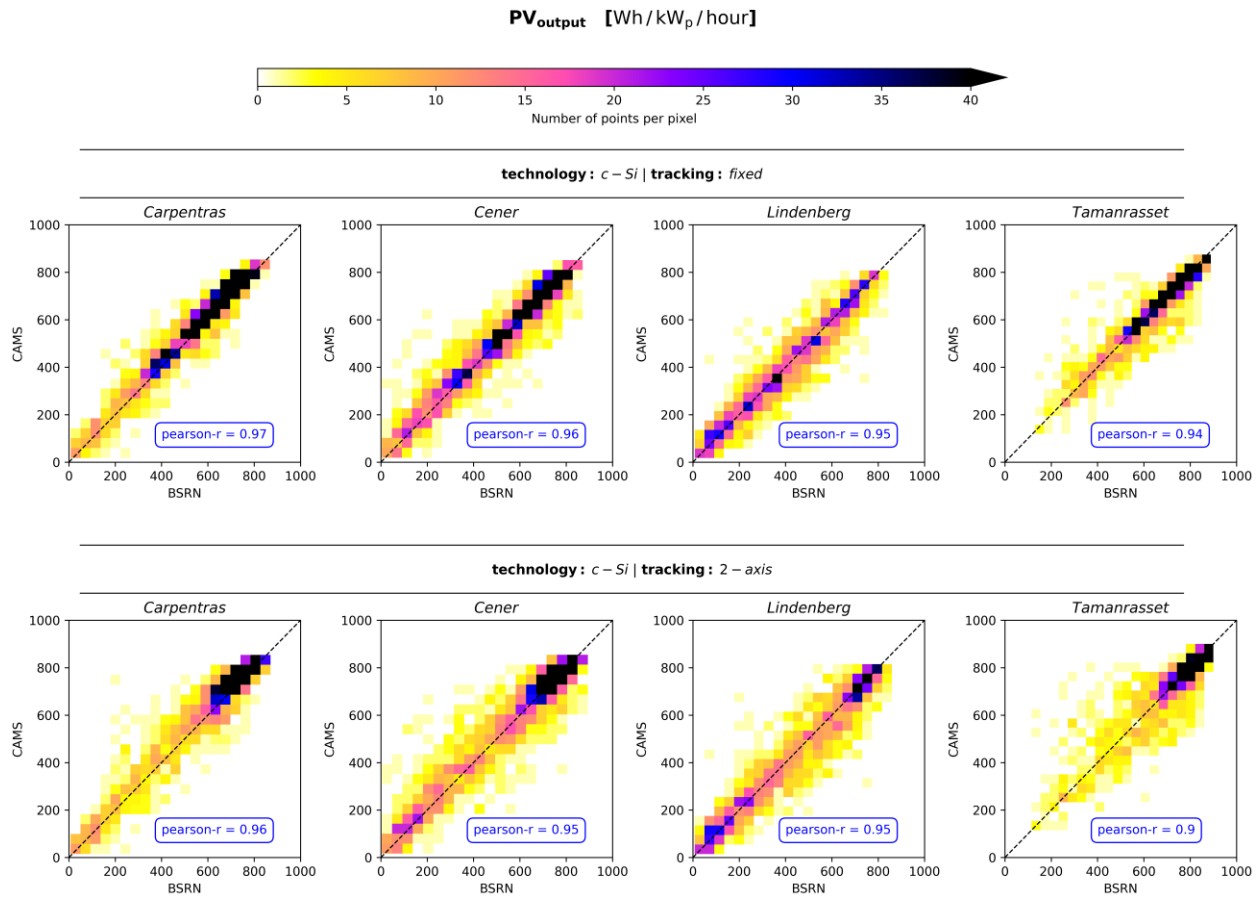

**Figure 10.** Overview of the reliability of the CAMS-based PV power simulations

## 4. Conclusions

The optimal approach to include solar radiation information to PV power models such as GSEE is to use actual in-situ measurements of global and diffuse solar irradiance. Since measurements of the diffuse component are rarely available, it is common to use measurements of the GHI (if available) and retrieve the diffuse component using a model such as BRL. In the absence of in-situ measurements, other options include the use of datasets such as CAMS or even a radiative transfer model, provided that atmospheric inputs such as clearness index, aerosol optical depth (AOD), and other aerosol properties are available. This study evaluated these options and their implications for PV modelling accuracy.

The results highlighted the importance of having precise information for the distribution of solar irradiance among its components in PV power modelling. The implementation of the BRL diffuse

fraction within GSEE serves as a practical, and under certain conditions, reliable solution to the
absence of detailed information for each component separately. Moreover, the integrated Climate
Data Interface submodule offers valuable prospects for investigating fluctuations in the solar PV
power generation across various timescales. In this context, the use of BRL has a key contribution
alongside the other computational procedures in processing climate datasets. Previous studies on
PV power modelling approaches have not examined their reliability under diverse atmospheric
conditions, including the effects associated with cloudiness, aerosol loading, as well as aerosol
optical properties.
The evaluation of the BRL's performance revealed a dependency of its reliability on the prevailing sky
conditions. BRL has excellent accuracy under totally clear sky scenes and still performs well for
cloudless scenes with moderate aerosol loading. In general, its accuracy is inversely proportional to
the complexity of the cloud scene. However, the model systematically underestimates the diffuse
fraction under high-loading conditions, such as during dust events. The discrepancies arising from
diffuse fraction estimation propagate to PV power generation and become particularly pronounced
in the modelling of 2-axis tracking systems. Indicatively, MAE under cloud-free scenes with moderate
aerosol loading, ranges between 2.2 to 6.6 Wh/kWp/hour for fixed panels and 4.7 to 15.0
Wh/kWp/hour for 2-axis tracking systems. Under partly cloudy conditions, where the cloud scene is
more complex, the MAE increases substantially, ranging from 12.4 to 25.8 Wh/kWp/hour for fixed
panels and from 23.5 to 55.1 Wh/kWp/hour for 2-axis tracking systems. Moreover, during intense dust
events, MAE can reach up to 49.2 Wh/kWp/hour in Tamanrasset, which is comparable to that
computed under partly cloudy conditions. Overall, the rMBE remains within the $\pm5\%$, with the
exception of a limited cases under overcast conditions. The same analysis applied to CdTe panels
yielded similar results, with minor differences.
Aiming to provide an indicative assessment of the financial impacts of the effect of desert dust
aerosols, we assume that the statistical indices calculated for Tamanrasset are representative of a
large-scale solar farm located in the Sahara region, with 500 MW installed PV capacity and systems
equipped with 2-axis solar tracking system. For this hypothetical solar farm, according to the value
of the Mean Absolute Error (MAE) on Table 4 for very high aerosol loading, we estimate that the
produced energy is $0.0492\,[kWh/kWp/hour] \times 500 \times 10^3\,[kWp] = 24600\,[kWh/hour]$
$\xrightarrow{supposing\ 12\ sunlight\ hours\ per\ day} \sim 295200\,[kWh/day]$ less than the expected from the PV power
simulations. According to the global average auction prices for selling produced energy back to the
grid in 2021 (IRENA, n.d.), the overestimations are equivalent to a financial loss of
$0.039\ [USD/kWh] \times 295200[kWh/day] \approx 11{,}500\ USD/day$. Therefore, site assessments that do
not correctly account for the distribution of surface solar irradiance in the sky under desert dust
aerosol conditions may overestimate financial performance and the annual financial deficit could be
accumulated to hundreds of thousands of US dollars per year.
Comparing the range of computed errors, we observe that the errors arising from employing CAMS
rather than using ground-based measurements, even when the diffuse fraction is not provided, are
higher across the overwhelming majority of the considered sky conditions. More specifically,
regarding the overall performance, MAE when using CAMS ranges between 33.7 and 46.1
Wh/kWp/hour, while with ground-based GHI measurements, MAE remains below 10 Wh/kWp/hour
within the modelling of systems with fixed panels and can reach up to 27.8 Wh/kWp/hour within the
modelling of 2-axis tracking systems. This outcome highlights the value of ground-based
measurements.
To sum up, achieving the highest quality PV power simulations necessitates high-quality, concurrent
measurements of solar irradiance components. In absence of this, the submodules included in the
GSEE package enable reliable simulations under the vast majority of prevailing sky conditions. CAMS
serves as a valuable data source for PV power modelling, but it cannot fully replace the precision and
reliability of using ground-based measurements. The integration of aerosol correction within the BRL
model opens new possibilities for further improvements in the modelling of solar energy systems. A
more comprehensive assessment would require measured PV output data; however, acquiring
simultaneous direct and diffuse irradiance measurements at the same location as the solar farms
remains challenging.

**Data availability**
The BSRN data are freely available on the BSRN web-page (https://bsrn.awi.de/). The AERONET
version 3 products are freely available from the AERONET website (https://aeronet.gsfc.nasa.gov/).
The CAMS radiation time-series are available from the Atmosphere Data Store
(https://ads.atmosphere.copernicus.eu ). The rest of the data used in this paper are available upon
request from the authors.
**Author Contributions**
Conceptualization: NP and IF; Data curation: NP and KP; Formal analysis: NP; Funding acquisition:
CZ; Investigation: NP; Methodology: NP, IF, SK, AK and AG; Project administration: CZ; Resources: SP,
KP and LD; Software: NP; Supervision: IF; Validation: NP, IF and SP; Visualization: NP; Writing –
original draft: NP; Writing – review & editing: all authors
**Funding**
This work has been supported by the action titled "Support for upgrading the operation of the
National Network for Climate Change (CLIMPACT II)", funded by the Public Investment Program of
Greece, General Secretary of Research and Technology/Ministry of Development and Investments.
Part of this work was also supported by the COST Action Harmonia (CA21119) supported by COST
(European Cooperation in Science and Technology). This work was partially funded by the
Copernicus Climate Change Service under contracts C3S2 _461-1_GR (Seasonal to decadal
predictions for national renewable energy management).
**Acknowledgments**
We thank the teams of the AERONET for ground measurements and maintenance, and CAMS for the
data production and distribution. We would like to thank the five site instrument operators and
technical staff of the BSRN network stations who made the ground-based measurements feasible.
A. Gkikas, J. Kapsomenakis, and C.S. Zerefos also acknowledge "CAMS2_82 Project: Evaluation and
Quality Control (EQC) of global products."

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
