# Peer review of "PV power modelling using solar radiation from ground-based"

_EGUsphere, 2025_

## Author Response (AR1)

**Reply to anonymous Referee #1**

We would like to thank the anonymous Referee #1 for his/her detailed and constructive review. The comments and suggestions were very helpful and significantly contributed to improving the clarity, robustness, and overall quality of the manuscript.

In the following, analytical replies are provided to each of the reviewer's comments. Reviewer's comments are written in bold font. Line numbers, when provided, refer to the new version with track changes.

**The manuscript "PV power modelling using solar radiation from ground-based measurements and CAMS: Assessing the diffuse component related uncertainties leveraging the Global Solar Energy Estimator (GSEE)" by Nikolaos Papadimitriou et al. focuses on the exploitation of the impact of the partitioning of the global horizontal irradiance (GHI) in its direct and diffuse components on the PV power production by simulations with the widely used Global Solar Energy Estimator (GSEE) model. The solar irradiance, air temperature, aerosol optical depth input sources are the BSRN and AERONET measurements from 5 sites in Europe, North Atlantic Ocean and Sahara desert and the CAMS model. The diffuse fraction (DF), being rarely measured except in a limited number of sites, is estimated within the GSEE through the logistic Boland-Ridley-Lauret (BRL) model, based on the clearness index as the main parameter. The effects of different cloud cover and aerosol optical depth on DF, and the corresponding impact on the simulation of power production is evaluated for fixed and 2-axis tracking PV systems based on c-Si and CdTe technology. The effects on different timescales are also explored. Finally, an assessment of the financial impacts deriving from evaluating the DF of desert dust from the BRL model compared to the measured GHI components, for a hypothetic PV solar farm around Tamanrasset is provided.**

**The main conclusion is that the best agreement in DF estimation with BRL model and ground-based measurements is for cloud-free and very load to moderate aerosol, i.e. for the simplest atmospheric conditions to be modelled. The worst situation for a reliable power production estimation is for partially cloudy skies. In sites impacted by high dust load, the BRL underestimated the DF and the power generation is overestimated.**

We would like to thank the reviewer for his/her time and comments.

**General comments**

While it is clear that the GSEE is widely used and the BRL model is optimized for both the Northern and Southern Hemispheres, I suspect it is not the best model for analyzing the effects of aerosols on the GHI partition, as the clearness index is mostly influenced by cloudiness and the reader is not informed about how the aerosols are accounted for. The authors should address this aspect, which is a key point in the development of the work.

as well as discuss similar works, if any, dealing with this topic.

Despite the large amount of calculations done in this work, the results are not valued by an adequate discussion, both in qualitative and quantitative terms. As a general comment I suggest quantifying the results in the text (Results and Conclusions sections) and not leaving the values only in the supplementary material.

The authors do not cite in the Introduction any previous paper dealing with the estimation of PV power generation under different cloudiness and/or aerosol load conditions, nor compare any of their results with previous works. If a similar work is not found in literature, this aspect, which increases the importance of this study, should be emphasized both in the introduction and in the conclusions.

Overall, I recommend a major revision of the key points before publication.

**Reply**

We acknowledge the limitations of the BRL model, especially regarding the effects of aerosols on GHI partition. However, it constitutes a key submodule of the GSEE library, as it is integrated within the internal processing chain of the climate data interface, a feature that plays a central role in the applicability of GSEE to climate-driven assessments. Some text has been added in the Introduction to clarify this (lines 84-86)

Moreover, following an extensive review of the relevant literature indicates that, despite the large number of studies discussing the role of aerosols and clouds on the amount of and the distribution of the solar irradiance that reaches the Earth's surface (Fountoulakis et al., 2021; Kosmopoulos et al., 2018; Papachristopoulou et al., 2022; Amiridis et al., 2024; Kosmopoulos et al., 2017; Calastrini et al., 2024; Kouklaki et al., 2023), there is a lack of studies addressing reliability of PV power simulations under diverse atmospheric conditions due to inaccuracies in the representation of the diffuse component in PV power models. Furthermore, we were not able to find any study examining the reliability of CAMS radiation data for PV power potential assessments. Some relevant text has been added both the Introduction and the Conclusions (lines 98-102 and lines 586-589)

In addition, the manuscript has been revised to include a more detailed quantitative discussion of the evaluation is Sections 3.3 – 3.5, and some tables have been transferred from the Supplement to the main text.

REFERENCES

Amiridis, V., Kazadzis, S., Gkikas, A., Voudouri, K. A., Kouklaki, D., Koukouli, M.-E., Garane, K., Georgoulias, A. K., Solomos, S., Varlas, G., Kampouri, A., Founda, D., Psiloglou, B. E., Katsafados, P., Papachristopoulou, K., Fountoulakis, I., Raptis, P.-I., Georgiou, T., Gialitaki, A., … Zerefos, C. (2024). Natural aerosols, gaseous precursors and their impacts in Greece: A review from the remote sensing perspective. Atmosphere, 15(7), 753. https://doi.org/10.3390/atmos15070753

Calastrini, F., Messeri, G., & Orlandi, A. (2024). Long-range mineral dust transport events in Mediterranean countries. Air, 2(4), 444–467. https://doi.org/10.3390/air2040026

Kosmopoulos, P. G., Kazadzis, S., Taylor, M., Athanasopoulou, E., Speyer, O., Raptis, P. I., Marinou, E., Proestakis, E., Solomos, S., Gerasopoulos, E., Amiridis, V., Bais, A., & Kontoes, C. (2017). Dust impact on surface solar irradiance assessed with model simulations, satellite observations and ground-based measurements. Atmospheric Measurement Techniques, 10(7), 2435–2453. https://doi.org/10.5194/amt-10-2435-2017

Kosmopoulos, P., Kazadzis, S., El-Askary, H., Taylor, M., Gkikas, A., Proestakis, E., Kontoes, C., & El-Khayat, M. (2018). Earth-Observation-based estimation and forecasting of particulate matter impact on solar energy in Egypt. Remote Sensing, 10(12), 1870. https://doi.org/10.3390/rs10121870

Kouklaki, D., Kazadzis, S., Raptis, I.-P., Papachristopoulou, K., Fountoulakis, I., & Eleftheratos, K. (2023). Photovoltaic spectral responsivity and efficiency under different aerosol conditions. Energies, 16(18), 6644. https://doi.org/10.3390/en16186644

Papachristopoulou, K., Fountoulakis, I., Gkikas, A., Kosmopoulos, P. G., Nastos, P. T., Hatzaki, M., & Kazadzis, S. (2022). 15-year analysis of direct effects of total and dust aerosols in solar radiation/energy over the Mediterranean Basin. Remote Sensing, 14(7), 1535. https://doi.org/10.3390/rs14071535

**Specific comments**

**Lines 67-72: since the use of an empirical model constitutes a relevant part of the work, I suggest to detail a little the description of these models and in particular the**

**description of the BRL model, for example by mentioning here the variables that are used to derive the DF.**

**Reply**

Some text added in order to highlight the innovative formulation besides the BRL model (lines 70-81)

**Line 71: some information about the BRL model should be provided, since the model is widely used in the paper. Is this the only model for DF estimation incorporated in the GSEE?**

**Reply**

GSEE includes only the BRL model. Although the user in its single usage (modelling a PV plant with hourly time-series data in a specified project location) has the option to use other diffuse fraction models (such as those included in pvlib), the climate data interface tool is designed for deploying exclusively the BRL implementation.

Some text added to clarify this (lines 127-131).

**Table 3 with the libRadtran input parameters is somehow unclear. The SZA input is "with step 90°": what do the authors mean? In addition, is the wavelength dependence of the surface albedo, SSA and gg accounted for? Finally, I suggest "integrated water vapor" instead of "water vapor".**

**Reply**

The reference to SZA input was a transcription error that was introduced during the manuscript revision. As the BRL model requires hourly input data at exact hourly timestamps for at least one full day, we used the libRadtran option to set as input datetime accompanied with the coordinates instead of SZA directly. After, in the analysis we computed the corresponding SZA values. In section 3.2, we stated that we chose the summer solstice as a representative day with sufficient number daylight hours.

Regarding the wavelength dependence of SSA and gg is not accounted for in the present analysis. In the present, this dependence is not explicitly accounted for, as the objective is not a fully spectrally resolved radiative transfer analysis, but rather to investigate the differences associated with some representative SSA values.

We accepted also the suggestion for adding the word Integrated before Water Vapor.

**Lines 290-295 and Figure 2: the comparison of the DF from the BSRN measurements and the BRL model is tricky. Do the differences for SZA>60° increase because of the difficulty of the model in estimating the DF for high SZAs? The bottom of Figure 2 shows that this happens for cloud-free conditions above 70° (not 60° SZA as said in the text) and not for all sites: Izana do not show the SZA dependence. It appears also for the cleanest conditions, i.e. for AOD500≤0.05.**

**In my opinion this deserves a little bit more investigation, instead of simply limiting the comparison to SZAs below 60°, as in Figure 3.**

**Reply**

We agree that the shift becomes observable within the range of 60-70 degrees. However, the specific point at which SZA starts to affect cannot be identified with precision. Therefore, a value of 60 degrees adopted as reference limit for practical reasons related to solar energy production applications, as above this value the low absolute irradiance levels contribute less to total energy yield. Moreover, we investigated this in section 3.2 in the sensitivity, where under clear sky conditions the shift arising closer to 70 degrees.

In Izana, there is the influence of altitude, where the levels of the diffuse irradiance are significantly lower.

The difficulty of the model in estimating the diffuse fraction for high SZA values may arise from the symmetry of a typical daily profile of the diffuse fraction and the hourly clearness, as the model requires full-day input data for hourly clearness. This is confirmed also by the sensitivity analysis, as the model has similar behavior.

Some clarifying text has been added.

**Line 326: under overcast conditions the BRL DF takes a range of values, approximately from 0.6 to 1, while the BSRN DF is close to 1. This means that even for homogeneous sky conditions, isotropic radiation the BRL model is not capable of providing reliable DF estimates. The authors could also refer to the 3D variability of cloud properties, whose effect cannot be accounted for by a model like BRL.**

**Reply**

The vast majority of cases where the BRL diffuse fraction is below 0.8 while the observed is close to 1 correspond to periods involving rapid transitions between partly cloudy and overcast skies, occurring either during the hour itself or immediately before or after it. These discrepancies can be mainly attributed to limitations of the DNI-based characterization methodology for cloudiness.

Moreover, the presence of aerosols can amplify these discrepancies. Even under cloudy conditions, aerosol may coexist and contribute, despite the difficulty to be measured and quantified.

Furthermore, a limited number of cases identified during intense dust events at Tamanrasset and Izana, where the reduction of DNI was so pronounced that the applied DNI-based criterion classified these conditions as overcast.

However, we did not further investigate these inconsistencies, as the energy production levels during such periods are very low.

Some clarifying text has been added (lines 327-337).

**Section 3.2: the dependence on latitude and altitude is not discussed, although presented in Figure 5.**

**The results show that for totally scattering aerosols (SSA=1) the BRL model underestimates the DF, while for absorbing aerosols (SSA=0.7) is overestimates the DF.**

**The authors may briefly discuss the results of the sensitivity analysis in terms of how the BRL logistic model treats the aerosol-radiation interactions. The model does not explicitly include the aerosol optical properties, but incorporates their effects into the clearness index, together with those exerted by the clouds, which are by no means larger. In this section, instead, the authors examine in detail how aerosols influence the partitioning of GHI into direct and diffuse components, and the results obtained with the BRL model are strongly biased, as expected. This is also a consequence of how the model was conceived, in particular of the data with which the relationship between the DF and the geometric, meteorological and atmospheric variables were determined.**

**Reply**

Some clarifying text added to discuss the effect of altitude and altitude, and to emphasize on the range of the discrepancies as an outcome of the effect of aerosol with different optical properties.

**Section 3.3: the authors should comment the results presenting the quantification of the differences in power production derived from using only GHI or GHI and DHI as GSEE model input. Not all sites and all atmospheric conditions have to be considered, but at least for two sites with different characteristics, such as Izana and Lindenberg, for fixed and 2-axis tracking systems.**

**Reply**

Some text has been added, as well as Tables with the computed indices for Carpentras and Tamanrasset (as representative locations) have been transferred from the supplement to the manuscript.

**Lines 492-494: the data gaps are in the GHI data and/or in the irradiance components? Can the author suggest a reason for these gaps? Does using a less stringent condition on the number of days per month (for example 15 days) allow for a less fragmented annual evolution?**

**Reply**

The data gaps occur mainly close to the solar noon during the summer months and arise from removing data through the quality control checks applied in the BSRN. Rejection of the data during the BSRN QC procedure is possibly related to operational issues at the station. Even though the data gaps are in most cases less than 2-3 hours, they may affect the BRL performance throughout these days. Thus, these days have been removed from the analysis.

**Lines 495-501: I find it very useful to present the differences also in percentage, referred to the energy production obtained using GHI and DHI as a reference.**

**Reply**

A Figure showing the percentage differences added in the supplement. Some discussion relative to the new figure has been added in the manuscript (lines 499-501).

**Section 3.5: more details about the CAMS data selection are needed. For example: the authors say that the CAMS solar radiation data are adapted to the investigated sites, but how is this done? By interpolation, by considering the nearest CAMS grid point? Moreover, the Izana site is excluded from the analysis because the altitude of the site is**

not directly comparable to the model vertical grid. For this reason, it is useful to know the CAMS 3D spatial grid and to add it in Section 2.3.

Another missing point is the quantification of the irradiance differences on power production, a qualitative discussion is not sufficient.

Finally, the authors should refer to previous papers, if any, dealing with the use of CAMS irradiance data for modelling PV potential power production.

**Reply**

For the data selection we used the "CAMS solar radiation time-series" product, where the user sets as input the coordinates as well as the altitude of a specific location, and then the output data are offered in ASCII format. Therefore, the interpolation methodology is integrated to the CAMS product algorithm. Some text added to clarify this in section 2.3.

Some text has also been added to discuss quantitively the validation as well as to highlight the novelty of this work, as we could not find studies directly examining CAMS performance for PV power modelling.

Conclusions: should summarize the main results and report some numerical data, which otherwise are only relegated in the supplementary material. For example, the results may be evaluated for tow or three sites with different characteristics in terms of latitude and aerosol regimes. I suggest reporting the impact on the power generation not only as absolute values, but also as percent, to facilitate understanding. In addition, even if the results pertaining to the panels with CdTe technology are not different from those obtained for c-Si panels, they should be briefly recalled.

When discussing the assessment of the financial impacts of the desert dust effects at Tamanrasset, the authors should clearly address that the analysis considers the differences in diffuse function derived from the measurements and calculated by the model. The sentence in lines 474-476 "Therefore, site assessments that do not account for the impact of desert dust aerosols may overestimate financial performance…." may be misleading, as it may be interpreted as the assumption that desert dust is not accounted for in the model simulations. I suggest reformulating the sentence.

**Reply**

We tried to improve the conclusions section as recommended by the reviewer.

**Technical corrections**

We thank the reviewer for the detailed comments and suggestions. All minor editorial, typographical, and wording-related comments (e.g., punctuation, wording clarifications, terminology corrections) have been carefully addressed and incorporated into the revised manuscript.

Responses to the comments requiring further clarification are provided below.

**Line 142: can the authors quantify how much is the uncertainty on using a fixed surface albedo of 0.3?**

**Reply**

The default GSEE value of 0.3 can be considered as representative for most types of surfaces (e.g., the surface albedo for urban landscapes, desert, and green grass is usually between 0.2 and 0.4). There are of course darker surfaces (e.g., forests, asphalt).

In the context of past studies (see reply to reviewer 1 in Papachristopoulou et al., 2024: https://doi.org/10.5194/amt-2023-110-AC1 ), we have investigated the sensitivity of the GHI irradiance to the surface albedo under various conditions (see the Figures below). For SZAs below 75° and under clear sky conditions, changing surface albedo by 10% changes the GHI by less than 1% (i.e., the difference is within the uncertainty range of the ground-based measurements). Under cloudy conditions the % differences are larger (i.e., ~ 5% for a 10% change in surface albedo for Cloud Optical Thickness of 12), but under such conditions the amount of PV power potential is small. Since very high surface albedo values are rare at the latitudes where the study is focused, and differences from the default value are generally smaller than 0.2, we decided to use the default surface albedo value. The manuscript has been also updated with this information (see lines 142-145)

**Line 159: is this reference correctly cited?**

**Reply**

Yes, the reference is correctly cited. It corresponds to an online source from the official GSEE model website, which has been cited accordingly in the revised manuscript.

**Table 1: is it necessary? In my opinion the text description is exhaustive.**

**Reply**

We have retained Table 1, as it provides a concise overview of the input parameters used in the GSEE model for the purposes of this study, complementing the textual description and improving clarity for the reader.

**Line 198: is it "daily" or "hourly"?**

**Reply**

It is daily, as this refers to the pre-processing of the data that used as input to the climate interface

**Lines 256-257: I suggest to include a sentence on how the solar radiation is estimated in CAMS and the 3D spatial resolution, an important information for the CAMS data selection operated for the analysis described in Section 3.5.**

**Reply**

Additional text has been added to clarify that the CAMS solar radiation time-series product is used. The description now briefly outlines how the data are retrieved, including the use of location coordinates and station altitude as inputs (lines 261-263).

**Line 367: maybe "ground altitude"?**

**Reply**

This was a typographical error, which has been corrected in the revised manuscript. The correct value is 0.1 km, rather than "0,1".

**Line 598: express the power overestimation also in percent number and the site where this is observed (should be Tamanrasset). Is the 49.2 Wh/kWp/hour on hourly value? I could not find this number in the supplementary material tables.**

**Reply**

This issue was caused by a typographical error and the inadvertent omission of several rows from the table, which have now been corrected in the revised manuscript. The value of 49.2

Wh/kWp per hour refers to an hourly estimate for the Tamanrasset site and is now correctly reported in Table 4. The relative mean bias error (rMBE) is expressed percentage terms.

**Line 613: Table 4 is not present.**

**Reply**

Table 4 had been removed during a previous revision and transferred to the Supplementary Material; however, the reference in the main text was not updated accordingly. In the revised manuscript, Table 4 has been reinstated in the main text and the reference has been corrected.

**Figure 6: bottom graphs. Why the data produced with only GHI for Izana seem to be cut around 900 Wh/kW$_p$ per hour?**

**Reply**

The PV systems considered in this study have a nominal capacity of 1 kWp. The PV model applies a default system loss factor of 10%. This effectively limits the maximum achievable power output to approximately 90% of the nominal capacity (i.e., around 900 W/kWp). This effect becomes apparent at the Izaña site due to its low latitude combined with its specific geographical and atmospheric conditions, which lead to high irradiance levels. As a result, the simulated PV output appears capped around 900 Wh/kWp per hour when only GHI is used.

**Table S2: the metrics for high and very high aerosol load are missing.**

**Reply**

The issue was due to missing rows in the previous version of Table S2. The table has been corrected and is now included as Table 4 in the revised manuscript, containing all metrics for low, high, and very high aerosol load conditions.

**Reply to anonymous Referee #2**

We would like to thank the anonymous Referee #2 for the careful evaluation of the manuscript and the helpful comment and suggestions. The reviewer's remarks were valuable in improving the presentation, consistency, and technical clarity of the study, and helped us to address several points that required further clarification.

Detailed responses to the reviewer's comments are provided below. The reviewer's comments are reported in bold font, and line numbers, when provided, refer to the revised manuscript with track changes enabled.

**This manuscript presents a comprehensive assessment of the impact of uncertainty in the diffuse component of solar radiation on the prediction of photovoltaic energy production using the Global Solar Energy Estimator (GSEE) and different data sources. This analysis includes categorization based on sky and aerosol conditions. The manuscript is well-organized and well-written, and the topic is worthy of study. However, there are some aspects that still need clarification or improvement.**

We would like to thank the reviewer for his/her time and comments.

**The specific comments are as follows:**

**1. Throughout the document, the existence of a reflected component of solar radiation on a tilted surface is not explicitly mentioned. It is understood that the authors include this component in the "diffuse" radiation, although, usually, in much literature, diffuse (sky) and reflected components are treated separately. Furthermore, the document does not specify which transposition model is used to determine solar irradiance on the inclined surface (in the same way that the separation model included in GSEE for estimating the diffuse fraction is mentioned). Several transposition models exist that treat diffuse sky irradiance in different ways, from the simplest, which considers it isotropic, to the more elaborate ones that separate the sky into different regions (background diffuse, circumsolar region, horizon brightening). This is important because fixed systems with different tilt angles and two-axis tracking systems are studied in different locations, and specifying which transposition model was used will allow for a better discussion and understanding of the results.**

**Reply**

The GSEE package includes the submodule "trigon", which contains a set of functions for computing the in-plane irradiance. These functions are based on trigonometric formulations, that account of the surface albedo, thereby including the ground-reflected component of solar radiation.

The main submodule of the GSEE library used to simulate the electric output of a PV panel requires as input the separated irradiance data (GHI, and diffuse fraction). Subsequently, it internally calls the trigon transposition model to compute the plane-of-array irradiance, which is then used to estimate the PV power output.

In contrast to BRL submodule – where apart from the climate data interface, alternative diffuse fraction models can be selected by the user in single-site application – the transposition model in GSEE is fixed and cannot be modified by the user.

Some text has been added (lines 133-137)

**2. Line 193 states that the data were "resampled to hourly", but this is unclear and needs clarification. Is an hourly average calculated? Refer also to Line 260 where hourly values are mentioned. The BRL model is based on irradiance values integrated over a one-hour period (Ridley et al., 2010). Please clarify. Regarding the AERONET data, how were the values treated given that the raw data does not have a defined periodicity?**

**Reply**

The resampling to hourly resolutions refers to the calculation of hourly mean values. The implementation of the BRL model included within the GSEE requires full-days timeseries of the hourly clearness index. For this purpose, the mean GHI of each hour is divided with the TOA on horizontal plane, which calculated as follows: Solar_Constant (Day_of_Year) * cos(SZA), where SZA is evaluated at the midpoint of the hour.

Regarding the AERONET data, although the original measurements do not have a fixed temporal periodicity, the data are resampled by computing hourly mean values from all available observations within each hour. This procedure ensures temporal consistency with the hourly irradiance and BRL model inputs.

The calculation of hourly mean values is clarified in the revised text.

**3. In Line 211: How the values of 0 < SV < 1 for partly cloudy conditions are determined? This is not clear. Also, if using a value of SV between 0 and 1 to**

characterize the intra-hour conditions then the value of irradiance must be hourly mean values (see also comment 2). Please clarify this aspect and justify the use of SV compared to the use of other indicators, such as the clear-sky index (not the clearness index), defined as the ratio of actual irradiance to irradiance for clear sky conditions from a suitable model, for example. This also relates to what is said in Line 290, where the use of cameras is suggested to overcome the issue of small and scattered clouds within the sky dome that enhances the diffuse component while not blocking the direct normal irradiance. There are other effective methods for identifying sky conditions. Please comment and clarify.

**Reply**

Representing the effect of cloudiness was challenging, as it requires the deployment of several observations. However, the DNI-based formulation aims to provide an indicative measure of the intra-hour cloudiness conditions. Alternative approaches, such as the clear-sky index or the Cloud Modification Factor require estimates of the clear sky GHI, which also introduce uncertainty. Some text added in lines 202 to 208.

**4. In Line 220, regarding the AERONET data, please clarify "… data were resampled at hourly intervals …" in view of comment 2 above.**

**Reply**

Some clarifying text has been added.

**5. In lines 234 and 247, there appear to be some typos (an extra "c" and "my", respectively).**

**Reply**

All the typos have been removed.

**6. In Figure 3: the represented data is only for SZA < 60, correct? Please, confirm this and mention it in the figure caption.**

**Reply**

Yes, we confirm that the figures is for SZA < 60. The figure caption revised to include this information.

**7. In Figure 8 and associated text, the reason why using a 30-days moving average is not clear. Is a centered moving average used? Please clarify, also regarding the data gaps which, up to this point in the document, were not evident (may be including in Table 2 this information will help).**

**Reply**

A centered 30-day moving average is used in Figure 8. The purpose of applying the moving average is to reduce short-term variability and highlight the underlying temporal behavior of the analyzed quantities, facilitating a clearer comparison of trends.

The data gaps occur mainly around solar noon and arise from measurements removed during the application of the BSRN quality control checks. Although these gaps are, in most cases, shorter than 2–3 hours, they may affect the BRL model performance for the corresponding days. This information has now been clarified in the revised text (lines 189-192).

[revised manuscript text omitted]
. Within the modelling of PV plants equipped with 2-axis solar tracking system, the deviations are much more pronounced relative to optimally inclined panels. BRL has excellent accuracy under totally clear sky scenes and still performs well for cloudless scenes with moderate aerosol loading. In general, its accuracy is inversely proportional to the complexity of the cloud scene. However, the model systematically underestimates the diffuse fraction under high-loading conditions, such as during dust events. Under such circumstances, this bias can potentially lead to significant overestimation of power generation by up to 49.2 Wh/kWp/hour 
[revised manuscript text omitted]

**Supplement**

 **Nomenclature**

| Acronym | Definition |
|---|---|
|  |  |
| AERONET | AErosol RObotic NETwork |
| AE | Angström Exponent |
| AOD | Aerosol Optical Depth |
| BRL | Boland-Ridley-Lauret diffuse fraction model |
| BSRN | Baseline Surface Radiation Network |
| CAMS | Copernicus Atmosphere Monitoring Service: ECMWF tool for atmospheric composition knowledge |
|  |  |
| DHI | Diffuse Horizontal Irradiance |
| DNI | Direct Normal Irradiance |
| DWD | Deutscher Wetterdienst: German Meteorological Service |
| ECMWF | European Centre for Medium-Range Weather Forecasts |
| FMF | Fine Mode Fraction |
| gg | Assymetry factor |
| GHI | Global Horizontal Irradiance |
| GSEE | Global Solar Energy Estimator |
|  |  |
| MAE | Man Absolute Error |
| MBE | Mean Bias Error |
|  |  |
| MOL-RAO | Meteorologisches Observatorium Lindenberg, Richard-Aßmann-Observatorium: DWD Observatory Lindenberg, Lindenberg (Tauche), Germany |
| NEO | Navarino Environmental Observatory, Messinia, Greece |

| OMN |  |
|---|---|
| PDFs | Probability Density Functions |
| PMOD/WRC | Physikalisch-Meteorologisches Observatorium Davos / World Radiation Center, Davos, Switzerland |
| PV | Photovoltaic |
| QC | Quality Check |
| r | Weighted correlation coefficient |
| rMBE | Relative Mean Bias Error |
| RMSE | Root Mean Square Error |
| RTM | Radiative Transfer Model |
| SDA | Spectral Deconvolution Algorithm |
| SSA | Single Scattering Albedo |
| SV | Solar Visibility |
| SZA | Solar Zenith Angle |
| TOC | Total Ozone Column (in DU) |
| TPM | Faculty of Technology, Policy, and Management, Delft, the Netherlands |
| UTC | Coordinated Universal Time |
| WMO | World Meteorological Organisation |

**Evaluation Metrics**

The formulas for the evaluation metrics used are the following:

1. Root Mean Square Error (RMSE)

$$RMSE = \sqrt{\frac{1}{N}\sum(x_{mod} - x_{obs})^2}$$

2. Mean Absolute Error (MAE)

$$MAE = \frac{1}{N}\sum|x_{mod} - x_{obs}|$$

3. relative Mean Bias Error (rMBE)

$$rMBE = \frac{1}{N}\sum\left(\frac{x_{mod} - x_{obs}}{x_{obs}}\right) \times 100\%$$

...

[Figure]

**Figure S1.** Difference between the diffuse fraction derived directly from the computations of DHI

and GHI using libRadtran and the one estimated by applying BRL to the libRadtran-computed GHI

for surface albedo 0.8

...

[Figure]

**Figure S2.** Comparison of the estimated hourly PV power generation between simulations performed using GSEE with input data consisting of either only GHI or both GHI and DHI under varying cloudiness conditions for panels with CdTe technology

[Figure]

**Figure S3.** Comparison of the estimated hourly PV power generation between simulations performed using GSEE with input data consisting of either only GHI or both GHI and DHI under varying aerosol conditions for panels with CdTe technology

...

| STATION: Carpentras | | fixed panels | | | 2-axis tracking | | |
|---|---|---|---|---|---|---|---|
| | |   |   |   |   |   |   |
|  | |  |  |  |  |  |  |
|  |  |  |  |  |  |  |  |
| |  |  |  |  |  |  |  |
| |  |  |  |  |  |  |  |
|  |  |  |  |  |  |  |  |
| |  |  |  |  |  |  |  |
| |  |  |  |  |  |  |  |
| |  |  |  |  |  |  |  |

**Table S2.**

| STATION: Tamanrasset | | fixed panels | | | 2-axis tracking | | |
|---|---|---|---|---|---|---|---|
| | | RMSE (Wh/kWp/hour) | MAE (Wh/kWp/hour) | rMBE (%) | RMSE (Wh/kWp/hour) | MAE (Wh/kWp/hour) | rMBE (%) |
| All-Sky scenes | | 13.6 | 9.3 | 1.0 | 40.4 | 27.8 | 3.8 |
| All-Sky scenes (cloudiness) | cloud-free | 11.5 | 8.0 | 0.8 | 35.3 | 23.4 | 2.9 |
| | partly cloudy | 20.1 | 15.0 | 2.0 | 56.1 | 45.7 | 8.1 |
| | cloudy (overcast) | 8.4 | 5.2 | -0.1 | 45.3 | 30.1 | 11.2 |
| Cloudless-Sky scenes (aerosol load) | low | 3.2 | 2.0 | 0.2 | 6.6 | 4.0 | 0.3 |
| | moderate | 5.4 | 4.6 | 0.6 | 13.0 | 10.5 | 1.2 |

**Table S1.** Evaluation metrics for GSEE performance within hourly intervals in Cener, comparing simulations with diffuse fraction from measurements and from the BRL model.

| STATION: Cener | | fixed panels | | | 2-axis tracking | | |
|---|---|---|---|---|---|---|---|
| | | RMSE (Wh/kWp/hour) | MAE (Wh/kWp/hour) | rMBE (%) | RMSE (Wh/kWp/hour) | MAE (Wh/kWp/hour) | rMBE (%) |
| All-Sky scenes | | 14.5 | 8.2 | 1.2 | 27.1 | 16.7 | 2.3 |
| All-Sky scenes (cloudiness) | cloud-free | 11.7 | 6.4 | 0.8 | 19.5 | 11.9 | 1.3 |
| | partly cloudy | 19.3 | 12.4 | 2.0 | 37.5 | 26.4 | 4.1 |
| | cloudy (overcast) | 4.7 | 2.7 | 1.6 | 11.2 | 6.3 | 4.6 |
| Cloudless-Sky scenes (aerosol load) | clear sky / low | 4.0 | 2.5 | -0.2 | 7.9 | 5.5 | -0.4 |
| | moderate | 6.9 | 3.1 | 0.4 | 11.4 | 6.2 | 0.6 |
| | high | 8.7 | 6.2 | 1.0 | 15.4 | 12.8 | 1.8 |
| | very high | NaN | NaN | NaN | NaN | NaN | NaN |

**Table S2.** Evaluation metrics for GSEE performance within hourly intervals in Lindenberg, comparing simulations with diffuse fraction from measurements and from the BRL model.

| STATION: Lindenberg | | fixed panels | | | 2-axis tracking | | |
|---|---|---|---|---|---|---|---|
| | | RMSE (Wh/kWp/hour) | MAE (Wh/kWp/hour) | rMBE (%) | RMSE (Wh/kWp/hour) | MAE (Wh/kWp/hour) | rMBE (%) |
| All-Sky scenes | | 16.0 | 9.7 | 1.8 | 26.1 | 17.0 | 2.7 |
| All-Sky scenes (cloudiness) | cloud-free | 11.8 | 6.7 | 0.9 | 20.7 | 13.5 | 1.4 |
| | partly cloudy | 20.4 | 13.9 | 2.3 | 32.4 | 23.5 | 3.5 |
| | cloudy (overcast) | 6.7 | 3.6 | 2.4 | 11.7 | 6.5 | 4.6 |

| Cloudless- | clear sky / low | NaN | NaN | NaN | NaN | NaN | NaN |
|---|---|---|---|---|---|---|---|
| Sky scenes | moderate | 8.9 | 5.5 | 0.5 | 14.9 | 9.7 | 0.6 |
| (aerosol | high | 12.6 | 10.3 | 1.4 | 19.5 | 15.9 | 2.0 |
| load) | very high | NaN | NaN | NaN | NaN | NaN | NaN |

**Table S3.** Evaluation metrics for GSEE performance within hourly intervals in Izana, comparing simulations with diffuse fraction from measurements and from the BRL model.

| STATION: Izana | | fixed panels | | | 2-axis tracking | | |
|---|---|---|---|---|---|---|---|
| | | RMSE (Wh/kWp/hour) | MAE (Wh/kWp/hour) | rMBE (%) | RMSE (Wh/kWp/hour) | MAE (Wh/kWp/hour) | rMBE (%) |
| All-Sky scenes | | 20.0 | 11.3 | 1.5 | 41.5 | 22.3 | 2.8 |
| All-Sky scenes (cloudiness) | cloud-free | 12.3 | 7.2 | 0.9 | 26.6 | 12.7 | 1.5 |
| | partly cloudy | 36.1 | 25.8 | 4.3 | 73.4 | 55.1 | 9.3 |
| | cloudy (overcast) | 16.8 | 11.8 | 4.6 | 35.5 | 26.0 | 11.8 |
| Cloudless-Sky scenes (aerosol load) | clear sky / low | 6.8 | 4.8 | 0.6 | 7.8 | 3.7 | 0.4 |
| | moderate | 9.3 | 6.5 | 0.9 | 20.8 | 15.0 | 1.8 |
| | high | 11.2 | 8.6 | 1.1 | 31.4 | 26.2 | 3.3 |
| | very high | 14.1 | 11.8 | 1.4 | 64.5 | 52.1 | 7.3 |

....

[Figure]

**Figure S4.** Percentage differences between the daily PV output estimated using the climate interface and the corresponding daily sums from hourly simulations as function of minimum daily

SZA and mean daily aerosol load for fixed panels

$\Delta$PVout $_{\text{climate interface, plant model}_{day}}$ **[%] = f(**min SZA $_{day}$**,** $\langle$AOD$_{500}\rangle_{day}$**)**

**technology :** $c - Si$
**tracking :** $2 - \text{axis tracking}$

**Figure S5.** Percentage differences between the daily PV output estimated using the climate interface and the corresponding daily sums from hourly simulations as function of minimum daily SZA and mean daily aerosol load for panels with 2-axis tracking

[Figure]

$\Delta$PVout $_{\text{climate interface, plant model}}$ **[%] = f(** Day of the Year **)** 〖kWh / kW$_p$ / day〗

**technology :** $c - Si$

**Figure S6.** Time-series of the percentage differences between the daily PV output estimated using the climate interface and the corresponding daily sums from hourly simulations

...

**Table** S4. Evaluation metrics assessing the reliability of GSEE Climate Data Interface in
estimating total daily PV output power in Carpentras

| STATION: Carpentras | | fixed panels | | | 2-axis tracking | | |
|---|---|---|---|---|---|---|---|
| | | RMSE (kWh/kWp/day) | MAE (kWh/kWp/day) | rMBE (%) | RMSE (kWh/kWp/day) | MAE (kWh/kWp/day) | rMBE (%) |
| All Days | | 0.22 | 0.17 | 1.7 | 1.24 | 1.08 | 15.5 |
| Sunny (cloudless) Days | | 0.19 | 0.15 | 1.8 | 1.19 | 1.08 | 14.1 |
| Sunny Days: average aerosol load | very-low aerosol | 0.18 | 0.15 | 2.1 | 0.82 | 0.72 | 10.0 |
| | aerosol-laden | 0.19 | 0.16 | 1.7 | 1.28 | 1.20 | 15.3 |

**Table** S5. Evaluation metrics assessing the reliability of GSEE Climate Data Interface in
estimating total daily PV output power in Tamanrasset

| STATION: Tamanrasset | | fixed panels | | | 2-axis tracking | | |
|---|---|---|---|---|---|---|---|
| | | RMSE (kWh/kWp/day) | MAE (kWh/kWp/day) | rMBE (%) | RMSE (kWh/kWp/day) | MAE (kWh/kWp/day) | rMBE (%) |
| All Days | | 0.24 | 0.22 | 3.4 | 1.45 | 1.34 | 19.0 |
| Sunny (cloudless) Days | | 0.22 | 0.20 | 3.4 | 1.20 | 1.11 | 14.4 |
| Sunny Days: average aerosol load | very-low aerosol | 0.15 | 0.14 | 2.5 | 0.66 | 0.62 | 7.7 |
| | aerosol-laden | 0.24 | 0.22 | 3.7 | 1.37 | 1.31 | 17.3 |

**Table** S6. Evaluation metrics assessing the reliability of GSEE Climate Data Interface in
estimating total daily PV output power in Cener

| STATION: Cener | | fixed panels | | | 2-axis tracking | | |
|---|---|---|---|---|---|---|---|
| | | RMSE (kWh/kWp/day) | MAE (kWh/kWp/day) | rMBE (%) | RMSE (kWh/kWp/day) | MAE (kWh/kWp/day) | rMBE (%) |
| All Days | | 0.24 | 0.18 | 2.5 | 1.28 | 1.08 | 16.8 |
| Sunny (cloudless) Days | | 0.26 | 0.21 | 3.3 | 1.15 | 1.00 | 13.6 |
| Sunny Days (average aerosol load) | aerosol-free | 0.18 | 0.15 | 2.4 | 0.73 | 0.60 | 7.6 |
| | aerosol-laden | 0.30 | 0.24 | 4.0 | 1.37 | 1.28 | 17.8 |

**Table** S7. Evaluation metrics assessing the reliability of GSEE Climate Data Interface in
estimating total daily PV output power in Lindenberg

| STATION: Lindenberg | | RMSE (kWh/kWp/day) | MAE (kWh/kWp/day) | rMBE (%) | RMSE (kWh/kWp/day) | MAE (kWh/kWp/day) | rMBE (%) |
|---|---|---|---|---|---|---|---|
| | | fixed panels | | | 2-axis tracking | | |
| All Days | | 0.24 | 0.18 | 2.1 | 0.99 | 0.81 | 15.2 |
| Sunny (cloudless) Days | | 0.29 | 0.23 | 4.4 | 1.04 | 0.96 | 14.6 |
| Sunny Days (average aerosol load) | aerosol-free | NaN | NaN | NaN | NaN | NaN | NaN |
| | aerosol-laden | NaN | NaN | NaN | NaN | NaN | NaN |

**Table S8.** Evaluation metrics assessing the reliability of GSEE Climate Data Interface in estimating total daily PV output power in Izana

| STATION: Izana | | fixed panels | | | 2-axis tracking | | |
|---|---|---|---|---|---|---|---|
| | | RMSE (kWh/kWp/day) | MAE (kWh/kWp/day) | rMBE (%) | RMSE (kWh/kWp/day) | MAE (kWh/kWp/day) | rMBE (%) |
| All Days | | 0.28 | 0.23 | 3.4 | 1.12 | 0.94 | 11.2 |
| Sunny (cloudless) Days | | 0.25 | 0.21 | 3.1 | 0.92 | 0.80 | 8.9 |
| Sunny Days (average aerosol load) | aerosol-free | 0.25 | 0.22 | 3.3 | 0.75 | 0.65 | 7.2 |
| | aerosol-laden | 0.24 | 0.19 | 2.3 | 1.38 | 1.31 | 15.3 |

...

**Table S9.** Evaluations metrics accessing the reliability of using CAMS solar radiation time- series for modelling PV output power in Carpentras

| STATION: Carpentras | | fixed panels | | | 2-axis tracking | | |
|---|---|---|---|---|---|---|---|
| | | RMSE (Wh/kWp/hour) | MAE (Wh/kWp/hour) | rMBE (%) | RMSE (Wh/kWp/hour) | MAE (Wh/kWp/hour) | rMBE (%) |
| All-Sky scenes | | 49.7 | 36.6 | 3.9 | 60.7 | 43.0 | 3.8 |
| All-Sky scenes (cloudiness) | cloud-free | 35.9 | 28.4 | 3.2 | 41.7 | 32.5 | 2.7 |
| | partly cloudy | 74.1 | 56.7 | 5.7 | 94.2 | 70.3 | 7.1 |
| | cloudy (overcast) | 46.7 | 37.4 | 11.4 | 49.2 | 38.3 | 13.9 |
| Cloudless-Sky scenes (aerosol load) | low | 25.0 | 20.0 | 2.1 | 28.8 | 22.3 | 1.5 |
| | moderate | 32.5 | 25.9 | 3.4 | 36.7 | 29.1 | 2.9 |
| | high | 41.8 | 36.0 | 5.0 | 49.0 | 41.3 | 4.7 |
| | very high | 42.3 | 36.9 | 6.1 | 49.9 | 44.1 | 6.3 |

**Table S12S10.** Evaluations metrics accessing the reliability of using CAMS solar radiation time-
series for modelling PV output power in Tamanrasset

| STATION: Tamanrasset | | fixed panels | | | 2-axis tracking | | |
|---|---|---|---|---|---|---|---|
| | | RMSE (Wh/kWp/hour) | MAE (Wh/kWp/hour) | rMBE (%) | RMSE (Wh/kWp/hour) | MAE (Wh/kWp/hour) | rMBE (%) |
| All-Sky scenes | | 55.9 | 33.7 | -1.0 | 75.2 | 46.1 | -0.6 |
| All-Sky scenes (cloudiness) | cloud-free | 32.4 | 20.8 | -0.7 | 45.2 | 29.8 | -0.7 |
| | partly cloudy | 87.3 | 67.5 | -4.4 | 111.8 | 86.9 | -3.0 |
| | cloudy (overcast) | 124.9 | 89.2 | 23.3 | 210.7 | 151.6 | 48.1 |
| Cloudless-Sky scenes (aerosol load) | low | 16.6 | 11.9 | 0.3 | 20.8 | 15.3 | -0.6 |
| | moderate | 19.7 | 14.9 | -0.4 | 28.2 | 20.9 | -1.2 |
| | high | 29.8 | 18.9 | -0.9 | 34.9 | 23.2 | -1.0 |
| | very high | 31.0 | 22.9 | 0.7 | 48.0 | 35.5 | 2.4 |

**Table S13S11.** Evaluations metrics accessing the reliability of using CAMS solar radiation time-
series for modelling PV output power in Cener

| STATION: Cener | | fixed panels | | | 2-axis tracking | | |
|---|---|---|---|---|---|---|---|
| | | RMSE (Wh/kWp/hour) | MAE (Wh/kWp/hour) | rMBE (%) | RMSE (Wh/kWp/hour) | MAE (Wh/kWp/hour) | rMBE (%) |
| All-Sky scenes | | 63.1 | 43.0 | 2.0 | 75.3 | 51.2 | 2.1 |
| All-Sky scenes (cloudiness) | cloud-free | 35.6 | 27.0 | 0.9 | 45.6 | 33.5 | 0.4 |
| | partly cloudy | 82.7 | 61.1 | 2.1 | 99.6 | 73.4 | 3.4 |
| | cloudy (overcast) | 77.3 | 50.8 | 20,7 | 83.5 | 53.4 | 24.8 |
| Cloudless-Sky scenes (aerosol load) | clear sky / low | 28.0 | 21.5 | 1.7 | 32.1 | 25.0 | 1.1 |
| | moderate | 38.5 | 30.4 | 2.8 | 48.3 | 38.8 | 2.3 |
| | high | 34.3 | 28.4 | 2.9 | 41.7 | 35.1 | 2.3 |
| | very high | NaN | NaN | NaN | NaN | NaN | NaN |

**Table S14S12.** Evaluations metrics accessing the reliability of using CAMS solar radiation time-
series for modelling PV output power in Lindenberg

| STATION: Lindenberg | | fixed panels | | | 2-axis tracking | | |
|---|---|---|---|---|---|---|---|
| | | RMSE (Wh/kWp/hour) | MAE (Wh/kWp/hour) | rMBE (%) | RMSE (Wh/kWp/hour) | MAE (Wh/kWp/hour) | rMBE (%) |
| All-Sky scenes | | 66.1 | 46.7 | -1.2 | 76.2 | 53.8 | -1.4 |
| | cloud-free | 37.9 | 24.9 | -1.7 | 50.7 | 33.4 | -2.5 |

| All-Sky scenes (cloudiness) | partly cloudy | 76.4 | 57.5 | -2.6 | 88.9 | 67.0 | -2.3 |
| | cloudy (overcast) | 60.9 | 42.3 | 8.0 | 63.9 | 43.2 | 9.5 |
| Cloudless-Sky scenes (aerosol load) | clear sky / low | NaN | NaN | NaN | NaN | NaN | NaN |
| | moderate | 42.3 | 28.7 | -2.5 | 53.4 | 24.7 | -3.0 |
| | high | 40.9 | 26.4 | -2.4 | 52.6 | 32.0 | -2.8 |
| | very high | NaN | NaN | NaN | NaN | NaN | NaN |